# Pronominal anaphora resolution in Polish: Investigating online sentence interpretation using eye-tracking

**Agata Wolna** \*, **Joanna Durlik, Zofia Wodniecka**

Psychology of Language and Bilingualism Lab, Institute of Psychology, Jagiellonian University, Krakow, Poland

* agata.wolna@doctoral.uj.edu.pl

**Data Availability Statement:** All data and the scripts necessary to reproduce the presented results are available from the Open Science Framework database (https://osf.io/dynwp/).

## Abstract

The mechanism of anaphora resolution is subject to large cross-linguistic differences. The most likely reason for this is the different sensitivity of pronouns to the range of factors that determine their reference. In the current study, we explored the mechanism of anaphora resolution in Polish. First, we explored preferences in the interpretation of null and overt pronouns in ambiguous sentences. More specifically, we investigated whether Polish speakers prefer to relate overt pronouns to subject or object antecedents. Subsequently, we tested the consequences of violating this bias when tracing the online sentence-interpretation process using eye-tracking. Our results show that Polish speakers have a strong preference for interpreting null pronouns as referring to subject antecedents and interpreting overt pronouns as referring to object antecedents. However, in online sentence interpretation, only overt pronouns showed sensitivity to a violation of the speaker's preference for a pronoun-antecedent match. This suggests that null pronoun resolution is more flexible than overt pronoun resolution. Our results indicate that it is much easier for Polish speakers to shift the reference of a null pronoun than an overt one whenever a pronoun is forced to refer to a less-preferred antecedent. These results are supported by naturalness ratings, which showed that null pronouns are considered equally natural regardless of their reference, while overt pronouns referring to subject antecedents are rated as considerably less natural than those referring to object antecedents. To explain this effect, we propose that the interpretation of null and overt pronouns is sensitive to different factors which determine their reference.

## Introduction

Understanding a sentence requires appropriate identification of what each word refers to.

For example, to fully comprehend the sentence (1):

(1) When Mary crossed the street, she looked back at the monument.

the reader needs to know that "she" in the second clause refers to "Mary" in the first clause and that both words are supposed to denote one and the same person. This sentence is an example of a pronominal anaphora, which is a particularly interesting instance of reference

**Funding:** This research was possible thanks to the grant SONATA BIS grant no 2015/18/E/HS6/00428 awarded to ZW by the National Science Center Poland (https://ncn.gov.pl/). During writing the paper JD was supported by Etiuda scholarship no 2018/28/T/HS6/00277 awarded by the Polish National Science Centre (https://ncn.gov.pl/). The funders had no role in study design, data collection and analysis, decision to publish, or preparation of the manuscript.

**Competing interests:** The authors have declared that no competing interests exist.

assignment. Understanding a pronoun requires a reader or listener to identify the appropriate referent from the grammatically correct and pragmatically possible antecedents. For example, in the following sentence (2):

(2) Mom waved to Mary when she was crossing the street.

both Mom and Mary are possible antecedents for the pronoun "she" in the subordinate clause. However, as shown by the example above, the pronouns themselves often do not provide enough information to identify the intended referent: the reference assignment depends mainly on the features of the antecedent itself. As a consequence, pronouns are inherently ambiguous: their reference depends on a range of different constraints, including morpho-syntactic, semantic, and discourse-related cues [1, 2]. In this paper, we report two experiments that explored the pronominal anaphora resolution mechanism in Polish, a Slavic language whose mechanism of anaphora resolution has not yet been empirically tested. First, we tested native Polish speakers' preference for anaphora resolution in ambiguous sentences. In the next step, we tested the robustness of this preference by testing how its violation influenced the online sentence-interpretation process.

## What determines the mechanism of pronominal anaphora resolution in different languages?

The mechanism of anaphora resolution can be influenced by a range of factors or cues which affect different stages of pronoun interpretation: from syntactico-semantic processing to constructing mental representations of discourse. However, the degree to which anaphora resolution relies on different factors varies between languages; in consequence, there are substantial cross-linguistic differences between typologically distant languages [3]. For example, anaphora resolution strategies differ between *discourse-oriented languages*, such as Korean, Chinese, or Japanese, and *sentence-oriented languages*, such as English, Italian, or Spanish [3]. To establish the referent of an anaphorical expression, the latter group of languages relies on morpho-syntactic, sentence-internal cues related primarily to subject-verb agreement. For example, in some languages with subject-verb agreement (like Spanish, Italian, or Polish), information about the gender and number of the predicate's referent is morphologically encoded into the verb form. In the following Polish sentence (3):

(3) Tata pomacha-ł Marys-i i Kas-i kiedy **one przechodzi-ły** przez ulicę.

Dad waived to Mary and Kate when they were crossing the street.

Dad.NOM.SG waved-1SG.M.PST Mary-DAT.SG and Kate-DAT.SG when they-3PL.F cross-3PL.F.PST the street.ACC.SG.

the verb's suffix ("przechodzi-ły") encodes both the grammatical gender (feminine) and the grammatical number (plural), both of which unambiguously relate the pronoun ("one" / "they") to the antecedent (Mary and Kate) from the preceding clause. A consequence of subject-verb agreement is that whenever a pronoun in the verb phrase refers to a less-preferred, non-topical, or unexpected antecedent, the speaker needs to adjust the interpretation of the sentence and re-identify the referent of the anaphorical expression [3].

In contrast to sentence-oriented languages, discourse-oriented languages which do not have a verbal agreement system, such as Japanese, Chinese, or Korean, can rely much less (or not at all) on morpho-syntactic constraints [4]. Therefore, other reference-determining factors that are derived from discourse become much more important for the correct resolution of an anaphoric expression [3, 4].

Different sensitivity to morpho-syntactic and discourse-related factors can account for differences between discourse- and sentence-oriented languages; however, even typologically close languages, such as Spanish and Italian (both of which are sentence-oriented languages),

can have different anaphora resolution preferences [5]. Several theories have been proposed to explain the mechanisms of how anaphorical expressions are interpreted. These theories provide a set of generalized rules that make it possible to formulate specific predictions for particular languages and identify potential sources of the differences between them. Below, we discuss some of the most prominent theories which explain the basis of the anaphora resolution mechanism.

### Anaphora resolution mechanism theories

Most anaphora resolution theories agree that the crucial factor for reference assignment is the salience or prominence of the referent (antecedent) itself: the more salient the referent is, the more reduced an anaphorical expression can be [6, 7]. According to this proposal, the most-reduced pronominal expression (the *null* pronoun) is bound to the most salient of the possible antecedents. On the other hand, less-reduced expressions, like *overt* pronouns, refer to less-prominent antecedents (see Fig 1, adapted from Kaiser & Trueswell [1]).

The crucial issue for theories of anaphora resolution is to identify the source of the salience of an antecedent and define the relationship between different types of anaphoric expressions and the antecedents they retrieve. Several factors have been suggested to influence the salience of antecedents. Kaiser and Trueswell [1] proposed that they can be divided into two categories: syntactico-semantic or discourse-related. The former group of constraints includes factors like subjecthood or thematic roles. Subjecthood refers to the syntactic position that an antecedent occupies in a sentence; it has been shown that syntactic subjects are considered more prominent than antecedents in other syntactic positions (e.g., [8]). Thematic roles describe the role that an antecedent plays with respect to the verb; it has been shown that agents are considered more prominent than patients of actions [9]. On the level of discourse, the salience of antecedents was argued to be driven by factors that influence the information structure, such as topicalization or focus. Topicalization predicts that discourse topics are more accessible than other possible antecedents [10, 11]. Focus predicts that entities which convey new or unexpected information are more prominent than other antecedents [12–15]. It has also been shown that there is a preference for the first-mentioned antecedent in a given sentence or utterance (e.g., [16]) or that pronouns prefer referents that occupy the same grammatical role in another clause–so-called *grammatical parallelism bias* [17]. However, a growing number of studies suggest that an antecedent's salience is in fact bound by multiple constraints or factors [13, 18].

To sum up, the existing literature suggests that pronoun resolution is driven by the salience of the possible antecedents; however, different theories indicate various sources of antecedents' salience. In the following section, we discuss some of the most important theories which address this question.

**Accessibility Theory.** The first theoretical proposal that describes the relationship between the salience of an antecedent and the types of anaphoric expressions (i.e., null, overt,

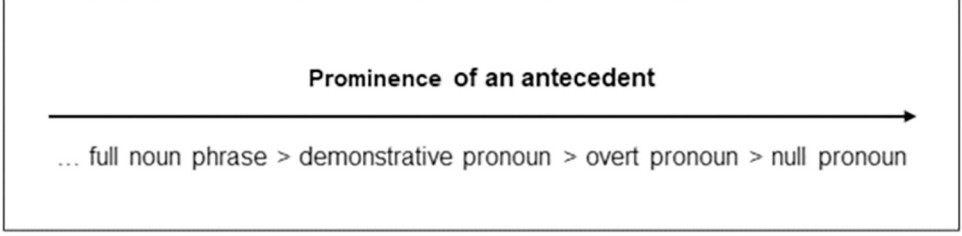

**Fig 1. Relationship between the prominence of an antecedent and the form of the pronominal expression that is necessary to refer to it: The more prominent the antecedent, the more reduced the pronominal form.**

stressed pronouns, etc.) is the Accessibility Theory (henceforth: AT [6]). It proposes that the salience of an antecedent depends on the degree of its accessibility in memory. AT proposes that the accessibility of an antecedent can be derived from three universal characteristics of the referring expression: informativity, rigidity, and attenuation [6]. The *informativity* of a given lexical marker predicts that an anaphorical pronoun is defined by the amount of semantic information it encodes. Therefore, the emptier a lexical marker is, the more informative the antecedent to which it refers. The *rigidity* of a lexical item describes how uniquely referring an expression is. This criterion can be context-sensitive, but AT argues that there are markers that are inherently more rigid than others (e.g., proper names like *Ernest Hemingway*, vs. personal pronouns like *he*). According to AT, more-rigid expressions tend to refer to less-prominent lexical items. The last dimension, *attenuation*, is related to the phonological size of the lexical marker: more-conspicuous phonological markers tend to refer to less-accessible antecedents (e.g., stressed vs. unstressed pronouns).

Summing up, AT predicts that *null* pronouns will refer to more-accessible and more-prominent antecedents than *overt* pronouns because null pronouns are less informative, less rigid, and more attenuated than overt ones. However, differences on the three dimensions of accessibility can give rise to differences between languages that are driven by language-specific sensitivity to different constraints on pronoun resolution.

**Position of the antecedent.** AT describes the relationship between the characteristics of a referring expression and the salience of the antecedent; however, it focuses on the properties of the referring expression itself. As such, AT does not propose any direct criteria for defining the salience of an antecedent. This issue has been addressed by the Position of Antecedent strategy (henceforth PoA [19]). Similar to AT, PoA assumes that anaphora resolution relies on the prominence of possible antecedents: the more prominent an antecedent is, the more reduced an anaphorical expression can be. PoA proposes that the accessibility of an antecedent depends on the universal syntactic roles of anaphorical expressions in sentences. According to PoA, a null pronoun should prefer the antecedent in a specific syntactic position: Spec IP (Specifier of the Inflectional Phrase). On the other hand, an overt pronoun would be more likely to relate to an antecedent that is not in a Spec IP position. Spec IP is a syntactically defined role in a sentence: it is occupied by an argument that defines an inflectional phrase which contains all information related to the tense and agreement features of a verb [20]. The precise type of the expression in a Spec IP position is further defined by the Extended Projection Principle (EPP), according to which the Spec IP of a given clause must be occupied by a syntactic subject [20], which in the case of Polish will always be a preverbal subject. Importantly, the EPP is not specific to any particular kind of clause: it is a general grammatical property of sentences. Therefore, the predictions of EPP should be generalizable across different languages [19]. As such, PoA refers to the universal syntactic properties of sentences to describe the anaphora resolution mechanism in different languages. It predicts that equivalent expressions (i.e., pronouns) in two languages should manifest the same interpretational bias when identifying the antecedent of the pronominal anaphora. This prediction is supported by experimental evidence in the case of the most-reduced pronominal form for a given language (e.g., a null pronoun in the case of Italian or Spanish, or a non-stressed pronoun in English). It has consistently been shown that the most-reduced pronoun relates to the antecedent in the Spec IP position in different languages, such as Italian [5, 19], Spanish [5, 21] or English [22]. However, the cross-linguistic consistency of anaphora resolution decreases when less-reduced overt (or stressed) pronouns are taken into account [5]. The inconsistency of the empirical findings suggests that–contrary to the PoA's assumption concerning the cross-linguistic universality of syntax-based rules for anaphora resolution–different languages can show varying sensitivity to syntactic cues of prominence.

**The Form-Specific Multiple-Constraint Approach.** One way of accounting for cross-linguistic variability that contradicts the syntax-based rules of reference assignment can be derived from a theory proposed by Kaiser and Trueswell [1]: the Form-Specific Multiple-Constraint Approach. According to this proposal, reference assignment takes context-dependent or discourse-dependent factors into account. Based on experimental evidence on anaphoric pronoun resolution in Finnish, Kaiser and Trueswell [1] showed that besides syntactic criteria such as those described by PoA, as well as mere word order, the prominence of an antecedent is also sensitive to the informational-structural factors encoded in the mental representation of a given discourse. Discourse representation includes information about the situation being described and the involved entities (such as people or objects–possible antecedents of a given clause). This information, which is inherently context-dependent, can influence the salience of antecedents and thus override syntactically encoded biases. According to Kaiser and Trueswell [1], referential expressions differ in their degree of sensitivity to the different types of cues that determine the prominence of an antecedent. What follows is that anaphoric expressions, both within and between languages, can assign different weights to the same types of cues. This approach provides a plausible explanation for the divergent pattern of overt pronoun interpretation that is observed not only between Italian and Spanish (5) but also between sentence-oriented and discourse-oriented languages, which use different types of cues to assign the reference to an anaphoric expression. For example, in the case of overt pronoun interpretation in Italian and Spanish, an increased processing cost resulting from forcing a shift of reference from the syntactic subject of a sentence (which is also the first-mentioned referent) to the syntactic object is observed for Italian but not for Spanish. Within the Form-Specific Multiple-Constraint Approach, this cross-linguistic difference can be explained by referring to the different sensitivity of pronouns to syntactico-semantic cues in both these languages. In Italian, pronoun resolution seems to rely more heavily on this type of cues than in Spanish. Similarly, differences between sentence-oriented and discourse-oriented languages can be accounted for by referring to the much higher sensitivity of pronouns to the informational structure of discourse in discourse-oriented languages than in sentence-oriented languages.

*Coherence-driven and centering-driven theories.* Among the anaphora resolution theories which argue that discourse-related factors are crucial in establishing the reference of a pronoun, the two most prominent proposals are the Coherence-driven approach [23, 24] and the Centering Theory (e.g., [25]). The former indicates the coherence of a discourse as a crucial notion in interpreting pronouns [23, 26]. According to Hobbs (e.g., [23]), the coherence of a discourse and the consequent pronoun interpretation is predominantly driven by semantics, causal interference, and world knowledge. Importantly, the coherence-driven accounts have long argued that constraints derived from the grammatical form or information structure are not necessary to explain the pronoun resolution mechanism. On the other hand, according to the Centering Theory [25], semantics and world knowledge do not play a role in pronoun resolution: it depends mostly on factors like grammatical roles in a sentence and information structure (e.g., word order, topic transitions, etc.). A recent proposal by Kehler and Rhode [24] attempts to reconcile these two accounts: it proposes that pronoun interpretation is determined, on the one hand, by probabilistic, coherence-driven expectations of what the following utterance will refer to; on the other hand, it is determined by constraints proposed by the Centering Theory, like grammatical role and information structure.

## Anaphora resolution in *pro*-drop languages: Experimental evidence

One way of testing a speaker's preference for pronominal anaphora resolution is to test the most-preferred interpretation of an ambiguous sentence in which the anaphorical pronoun

might refer to at least two different antecedents [17, 19, 27]. In ambiguous sentences in languages like Spanish [28] and Italian [19], both subject and object antecedents are grammatically coherent. The preference for anaphora resolution in ambiguous sentences usually follows the predictions of PoA: the null pronoun usually relates to the antecedent that is in a Spec IP position (i.e., the subject of the main clause), while the overt pronoun refers to the antecedent in a non-Spec IP position (i.e., the object of the main clause).

The anaphora resolution mechanism can also be explored by analyzing the consequences of violating the preferred pronoun-antecedent match. This refers to situations in which a pronoun is forced to co-refer with an antecedent that does not match its resolution preference. This violation forces the reader to shift the reference and override the bias for pronoun interpretation. As previously discussed, the interpretational bias might be a syntactically based preference for interpreting verbs as referring to the subject antecedent [1, 19, 29]. Still, it can also be driven by discourse-related factors like the preference for the first-mentioned antecedent (e.g., [16, 29]), for the topic [10, 12], for the focused entity [10, 11, 14], or for the agent [9] of the sentence. Regardless of the source of the preference for pronoun interpretation, the process of reference-shifting incurs cognitive effort, which can be observed in behavioral and eye-tracking measures as an increase in reading times (self-paced reading: [5]) or fixation times (eye-tracking: [21, 30]).

In a self-paced reading experiment, Filiaci and colleagues [5] tested native Italian and Spanish speakers. Participants were asked to read unambiguous sentences which either did or did not match the preferred anaphora interpretation. While both languages showed consistent processing costs (also referred to as *processing penalty*) for the null pronoun that was forced to refer to the object antecedent, the pattern of results showed cross-linguistic differences for the overt pronoun. In Italian, the interpretation of an overt pronoun that was forced to refer to a subject antecedent resulted in a significant processing penalty. This was not the case for native Spanish speakers, who could accommodate the less-preferred pronoun-antecedent match effortlessly. Notably, the effects of the overt pronoun reference shift from the preferred object antecedent to the less-preferred subject antecedent were observed only in the late part of sentences: the processing penalty was observed only for wrap-up parts of sentences that followed actual anaphorical expressions [5].

The consequences of violating the offline preference for pronominal anaphora resolution have also been explored using eye-tracking [21]. In a sentence-reading experiment with Spanish speakers, no processing penalty was observed for sentences that violated the null pronouns' referential preference. At the same time, the additional processing cost was pronounced for overt pronouns: fixation times were much longer for pronominal verb phrases (VPs) in which a pronoun referred to a subject antecedent than for pronominal verb phrases in which a pronoun referred to an object antecedent [21]. This pattern of results stands in contrast to the findings of a self-paced reading experiment carried out by Filiaci and colleagues [5], which showed the opposite effect in Italian: violation of a null pronoun's referential preference was related to a much larger processing penalty than a violation of an overt pronoun's referential preference. However, no differences in the sensitivity of null and overt pronouns to the violation of their preferences was observed for Spanish. Differences in whether the processing penalty is observed for sentences with null or overt pronouns referring to their less-preferred antecedents can be accounted for by design differences between the studies of Chamorro and colleagues and of Filiaci and colleagues. In particular, they can be accounted for by differences in the clause order: the stimuli in Chamorro and colleagues' study [21] followed the Main-Subordinate order, while in Filiaci and colleagues' study [5] the order was Subordinate-Main (for further discussion, see *Discussion*). Interestingly, despite the discrepancies between the results of these two experiments, the results of the experiment by Filiaci and colleagues [5]

demonstrate that the resolution preference for a null pronoun is similar for Italian and Spanish, while differences emerge for overt pronouns.

To conclude, the cross-linguistic differences in anaphora resolution might stem from the divergent sensitivity of null and overt pronouns to factors determining the reference of antecedents, which can be driven by syntactico-semantic or discourse-related constraints [1, 24]. Therefore, to provide an accurate description of pronominal anaphora resolution in a given language, experimental evidence is needed to assess differences in the sensitivity of pronouns to syntactico-semantic and discourse-related factors. The current paper aims to provide an empirically based description of pronominal anaphora resolution in Polish.

## Pronouns in the Polish language: Introductory remarks

Before discussing the details of our research questions, a couple of introductory remarks regarding the Polish language seem essential. Polish is a Slavic language whose dominant sentence structure is SVO (subject–verb–object); however, Polish is characterized by a relatively free word order. Polish is a morphologically rich language with two number classes, three gender markings, and a declension with seven cases reflected in the morphological form of nouns, adjectives, and some pronouns [31]. Verbs in Polish are also highly inflected and, inter alia, categories such as person, number, and gender are marked by almost every verb form. Nouns, pronouns, and verbs have to agree in person, number, and gender. Polish is also a *pro*-drop language that has a strong tendency to omit pre-verbal pronouns, which are redundant in most cases as the morphological form of a verb encodes the information that allows to identify the referent of a verb. In this aspect, Polish is similar to Romance languages such as Spanish or Italian, although the families of Slavic and Romance languages are generally typologically distant. To the best of our knowledge, neither the preferences for anaphora resolution in Polish speakers nor the influence of these preferences on online sentence interpretation have yet been empirically tested. The only known empirical report on Polish speakers' preference for pronominal anaphora resolution was published by Carminati [19], but it is based on data from only two participants. Carminati asked two Polish speakers to provide their preferred interpretation of ambiguous sentences containing a null or overt pronoun; she found that the null pronoun sentence was interpreted as referring to the subject antecedent, while the overt pronoun was assigned to the object antecedent. However, given the limited participant sample, these data cannot be treated as fully informative and reliable.

## The current research

The primary aim of the two experiments presented in the current paper was to empirically test the pattern of anaphora resolution in Polish. As previously discussed, the anaphora resolution pattern is bound to the notion of the prominence of an antecedent. Previous studies have shown that the grammatical role of the antecedent (especially the subjecthood) allows accurate predictions of how the pronominal anaphora will be interpreted. However, these predictions tend to be valid only for the most-reduced pronominal form allowed by a given language (which is usually a null or a non-stressed pronoun [5, 19, 22]). Less-reduced anaphorical forms (i.e., overt pronouns in *pro*-drop languages, stressed pronouns, or noun phrases) can be prone to the influence of other factors that determine the prominence of the antecedent. These factors can lead to vast cross-linguistic differences in anaphora resolution preference (e.g., [5]). Therefore, to provide an accurate description of the pronominal anaphora resolution preference in Polish, the predictions derived from the theoretical frameworks needed to be experimentally validated.

To this aim, we conducted two experiments. In the first one, we established the preference for anaphora resolution in native Polish speakers, who we asked to interpret ambiguous

sentences and rate their naturalness. Each sentence contained either a null or an overt pronoun in the subordinate clause that might refer to either the subject or the object antecedent introduced in the preceding ambiguous main clause (for examples of the sentences used in the experiment, see *Methods*). To the best of our knowledge, no empirical study (besides the anecdotal evidence provided by Carminati [19]) has yet addressed the question of the pronominal anaphora resolution pattern in Polish.

In the second experiment, we examined how the offline preference for anaphora resolution in Polish influenced the online sentence-interpretation process. We used eye-tracking to follow the time course of the interpretation of sentences containing null and overt pronouns that referred to either matched or unmatched antecedents. Following the design proposed by Chamorro and colleagues [21], we manipulated the grammatical number of a subject and object antecedent so that only one interpretation of the main clause containing a singular verb would be grammatically correct (for examples, see *Methods*). Analysis of the online sentence-interpretation process using an eye-tracker allowed us to assess the sensitivity of Polish pronouns to a violation of the syntax-based pattern of anaphora resolution preference. Additionally, participants were asked to rate the naturalness of the sentences they read.

## Experiment 1 –What is the offline pattern of anaphora resolution in native Polish speakers?

The first experiment aimed to explore the preference of native Polish speakers to interpret sentences that contain an overt or null pronoun. We tested if the presence of a pronoun influences the interpretation of a subordinate clause as referring to either the subject or the object of the main clause.

Specific predictions regarding the pronoun interpretation preference in ambiguous sentences in Polish could be formulated on the basis of the available theoretical proposals or empirical evidence from experiments on other *pro*-drop languages. The theoretical accounts of pronominal anaphora resolution predict that more-reduced pronouns refer to the most prominent of the possible antecedents. The syntax-based criterion of prominence proposed, e.g., by PoA, indicates that the most-salient antecedent is the subject of the main clause in the Spec IP position [19]. A similar prediction can be formulated based on the advantage of the first-mentioned antecedent [16, 29] or the fact that the first-mentioned antecedent is typically interpreted as the topic of a given sentence [32]. Therefore, we expected that the null pronoun of the subordinate clause would be interpreted as referring to the subject of the main clause. In contrast, we expected that the overt pronoun of the subordinate clause would be interpreted as referring to the object of the main clause. Further support for our prediction stems from the fact that a similar pattern of anaphora resolution in ambiguous sentences is shown in *pro*-drop Romance languages [5, 19, 28]. In Polish, the sentence structure is analogous to that of Italian or Spanish: the antecedent in a Spec IP position is always the preverbal subject. Moreover, following the studies of Alonso-Ovalle and colleagues [28] and Chamorro and colleagues [21], we expected to find that native Polish speakers would interpret *null* pronoun sentences as more natural than *overt* pronoun sentences. Raw data and the script necessary to reproduce the statistical analysis reported in Experiment 1 are freely available online: https://osf.io/dynwp/.

### Materials and methods

**Participants.**  Seventy native Polish speakers (8 males, age: M = 20,99; sd = 2,61) took part in the experiment. They were recruited from among students of Jagiellonian University and received course credits for their participation. Participants provided informed consent for

participation in the experiment in an online form (in which there was an option to consent to participate in the study or leave the experiment) before they started the experiment. After completing the survey, they were provided with a short explanation of the aim of the study. The experiment met the requirements of the Ethics Committee of the Institute of Psychology of Jagiellonian University concerning experimental studies with human subjects.

**Stimuli.** To test whether the pronoun form (null vs. overt) can influence its interpretation preference, we created 48 ambiguous sentences in Polish that allowed interpretation of a subordinate clause as referring to either the subject or the object of the main clause. Each sentence consisted of a main clause containing subject and object antecedents of the same number and gender, followed by a temporal adjunct subordinate clause (henceforth referred to as a *subordinate clause* introduced by the conjunction *kiedy*–"when"). The conjugation form of the verb in the subordinate clause matched the gender and number of the subject and object antecedents of the main clause, thus allowing ambiguous interpretation of the sentences. Two versions of each sentence were created that corresponded to two different conditions: the verb of the subordinate clause was preceded by either an overt or a null pronoun (both are allowed according to the morpho-syntactic rules of Polish). Each experimental stimuli followed the same sentence structure:

subject / verb / object / time conjunction ("when") / pronoun (null or overt) / verb / complement

By manipulating the *Pronoun*, two different conditions of each sentence were created (4–5):

Condition 1: **null pronoun**

(4) Mama   pomacha-ła   córce,   kiedy Ø   przechodzi-ła   przez ulicę.

The mother   waved to the   daughter   when Ø (she) was crossing   the street.

Mother.NOM.SG   waved-1SG.F.PST   daughter.DAT.SG   when   Ø (she)   cross-1SG.F. PST   street.ACC.SG.

Condition 2: **overt pronoun**

(5) Mama   pomachał-a   córce,   kiedy   ona   przechodzi-ła   przez ulicę.

The mother   waved to the   daughter   when she   was crossing   the street.

Mother.NOM.SG   waved-1SG-F-PST   daughter.DAT.SG   when   she   cross-1SG.F.PST   the street.ACC.SG.

**Procedure.** The experiment was conducted via an online survey platform (Qualtrics: www.qualtrics.com). It was preceded by written instructions (containing comprehension questions which made it possible to confirm that participants had read the instructions carefully) and a short training session. After completing the training, participants were allowed to re-read the instructions in case they had doubts regarding the experimental procedure.

Participants were instructed to read each sentence and proceed to the next screen, which contained a disambiguation question asking for their interpretation of the sentence they just read (e.g., "Who crossed the street?"). The participants were provided with two possible answers: one always referred to the subject of the main clause, while the other always referred to the object of the main clause. Participants made their decisions by clicking the corresponding button. After progressing to the next screen, they were asked to rate the naturalness of the sentence on a scale from 0 (completely unnatural) to 5 (perfectly natural). After rating the naturalness, they proceeded to the following sentence.

Each participant read 48 experimental sentences in one of the two conditions: null or overt pronoun. There were two versions of the task, each of which contained different variants of each sentence (null or overt), which were counterbalanced between participants. Additionally, the same number of filler sentences was presented. Filler sentences were of a similar form to the experimental sentences, but they contained different conjunctions (e.g., "before", "albeit", "since", etc.).

**Data analysis.** The preference for sentence interpretation was analyzed in R [33] using a general linear mixed-effects model (glmer) of a binominal family, as implemented in the lme4 package (version 1.1.26; [34]). We fitted a model predicting the preferred antecedent (subject or object) based on a categorical predictor *Pronoun* (null or overt), which was dummy coded prior to running the analysis. Random intercept and slope by participants and random intercept by items were included in the model. Naturalness ratings were analyzed using a cumulative ordinal model with flexible thresholds created in the Stan computational framework (http://mc-stan.org/) and implemented in the brms package (version 2.15.0; [35]). The model included the main effect of *Pronoun* (null or overt) as well as the main effect of *Antecedent* (subject or object), which referred to the participant's interpretation of a given sentence. The interaction between the effects of *Pronoun* and *Antecedent* was also included in the model. Random intercept and slopes for *Pronoun* and *Antecedent* by participants and items were included in the model. It has recently been shown that Bayesian-based ordinal models are a better choice for rating scale analyses as they can model responses as discrete and not necessarily equidistant data points. As such, they provide a much better fit for the data obtained from any type of rating scales [35, 36]. The effect sizes (Cohen's *d* statistic) were calculated using the lme.dscore function from the EMAtools package (version 0.1.3; [37]).

## Results

**Interpretation of the sentences.** In this experiment we aimed to determine if the preference for interpreting ambiguous sentences in Polish was modulated by the nature of the pronoun. We found a significant difference between the two *Pronoun* conditions ($z = -15.608$, $p > 0.001$, $d = -5.58$, see Fig 2). The results revealed that the preferred interpretation of the sentences strongly depends on the pronoun type: sentences containing a null pronoun were interpreted as matching the subject antecedent (percentage of subject-match answers: M = 79.7%, SE = 4.81%), while sentences containing an overt pronoun were interpreted as matching the object antecedent (percentage of object-match answers: M = 90.8%, SE = 3.45%).

**The naturalness of sentences.** Apart from examining Polish speakers' preference when interpreting ambiguous sentences, we also wanted to determine whether they judge sentences as more natural depending on the use of a pronoun. Participants rated the naturalness of each sentence on a scale from 0 to 5. We found a strong effect of *Pronoun* ($\beta = -0.44$, 95% CrI [-0.63–0.26]), thus indicating that native Polish speakers find sentences containing a null pronoun more natural than sentences containing an overt pronoun. No main effect of *Antecedent* was observed ($\beta = -0.07$, 95% CrI [-0.27 0.14]); however, we found a significant interaction between *Pronoun* and *Antecedent* ($\beta = -0.98$, 95% CrI [-1.24–0.73]). Direct comparisons between the two *Antecedent* conditions (i.e., subject- and object-matching sentences) revealed significant effects for sentences containing a null pronoun ($\beta = -0.42$, 95% HPD [-0.62–0.20]): subject-matching sentences were rated as more natural than object-matching sentences. Significant effects were also found for sentences containing overt pronouns ($\beta = 0.56$, 95% HPD [0.29 0.81]): in these cases, subject-matching sentences were rated as less natural than object-matching sentences. The results of the analysis and the mean scores for all conditions are presented in Table 1 and Fig 3.

**Experiment 1 –Summary.** The results of Experiment 1 showed that native Polish speakers prefer to interpret the null pronoun of a subordinate clause as referring to the subject of the main clause; they also prefer to interpret the overt pronoun of a subordinate clause as referring to the object of the main clause. In the subsequent experiment, we investigated the online processing of pronominal anaphoric structures in Polish and assessed the extent to which it is influenced by the preference for anaphora resolution established in Experiment 1.

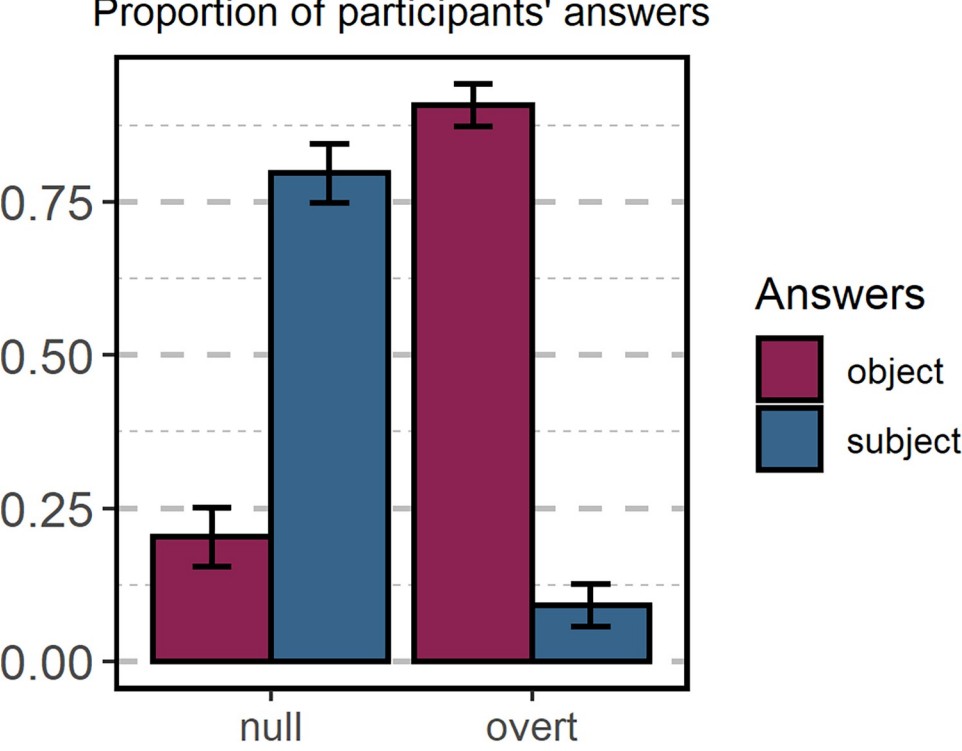

**Fig 2. Preference for sentence interpretation.** The proportion of subject- and object-match answers for sentences containing a null or overt pronoun. Error bars represent the standard error of the model's predictions.

### Experiment 2 –Eye-tracking measures of anaphora resolution in Polish

Based on experimental evidence from studies on Italian and Spanish [5, 19, 28], we expected that online pronoun resolution in Polish would be driven by the prominence of the available antecedents. As such, the interpretation of a null pronoun would refer to the most-prominent antecedent. In our experiment, it was always the first-mentioned antecedent that occupied the syntactic position of a subject. On the other hand, we expected that the interpretation of the overt pronoun would refer to the antecedent in the less-prominent syntactic position of the object, which is also the second-mentioned antecedent in a sentence and is a less likely topic of a given utterance. As discussed in the Introduction, violation of the preferred anaphora inter-pretation can result in a *processing penalty*, i.e., increased cognitive cost related to the reference

**Table 1. Mean scores and SD for the naturalness judgment of the experimental stimuli.**

| Pronoun | Interpretation (Antecedent) | Naturalness rating |
|---------|------------------------------|---------------------|
| **Null** | **Subject** | 3.69 *(1.26)* |
|  | **Object** | 3.14 *(1.32)* |
| **Overt** | **Subject** | 2.62 *(1.52)* |
|  | **Object** | 3.31 *(1.31)* |

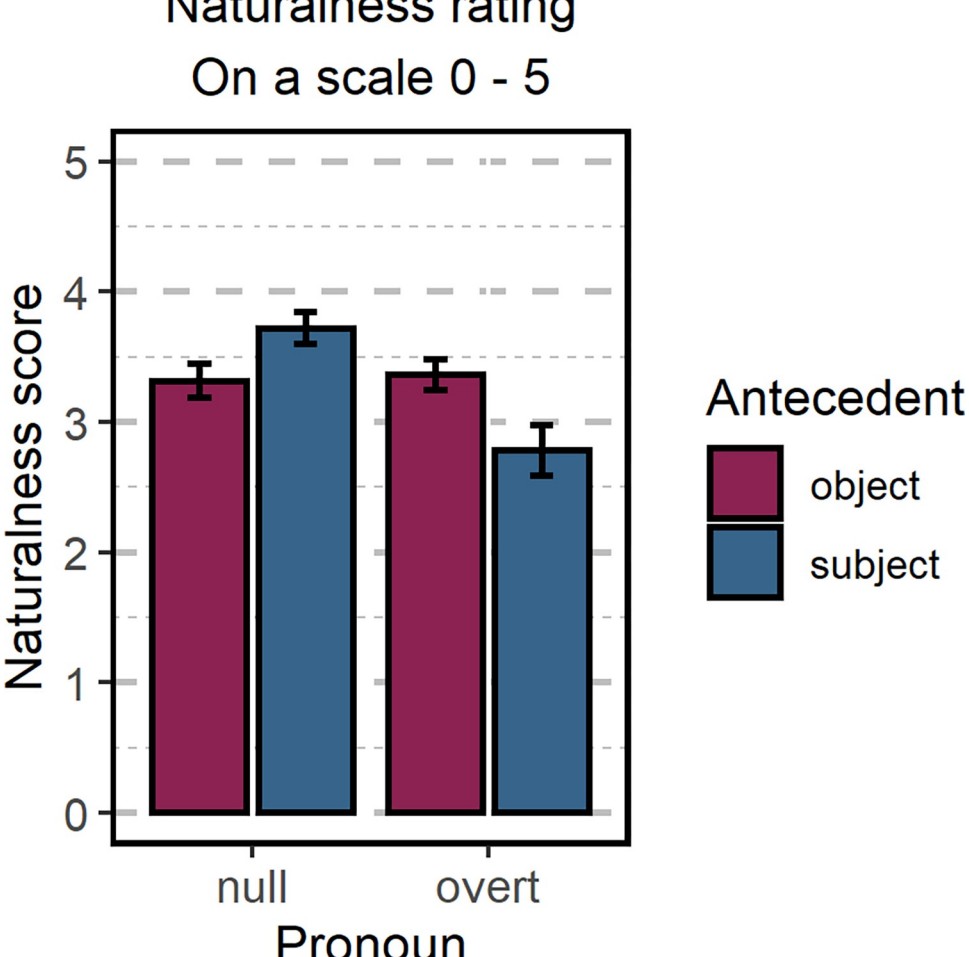

**Fig 3. Naturalness judgment of Experiment 1.** The naturalness rating of the sentences collected in Experiment 1. Error bars represent the standard error of the model's predictions.

shift. In eye-tracking, this cost should be reflected in longer reading times (measured by *first-pass time*, *go-past time*, and *total time*) in the critical region containing the VP of the subordinate clause. The eye-tracking experiment's critical manipulation involved forcing the pronoun to refer to the mismatching antecedent (i.e., a null pronoun matched with an object antecedent or an overt pronoun matched with a subject antecedent). This manipulation should allow us to observe the reader's reaction to a violation of the anaphora resolution preference.

Our experiment closely followed the design of the study by Chamorro and colleagues [21]. While many previous experiments used eye-tracking to assess pronoun resolution strategies, they used methodology that does not allow direct testing of how anaphora resolution preference modulates the unfolding sentence interpretation. Previous eye-tracking studies either used a visual-word paradigm (e.g., [1, 29, 38–40]) or if they were based on sentence-reading paradigms, they were designed to test different research questions than ours, e.g., questions related to the resolution of intra-sentential anaphora (processing costs for using "they" as a singular pronoun in English [41]; pronoun resolution bias related to the verb's implicit causality [42]; or the influence of cleft focus on pronoun accessibility [14]). Eye-tracking has also been used to assess the effect of immersion in a second-language environment on pronoun

resolution [21, 30]. One of these studies is the experiment by Chamorro and colleagues [21], which, to the best of our knowledge, is also the only experiment that used eye-tracking to assess the sensitivity of pronominal anaphora resolution to a violation of the preference for a pronoun-antecedent match. Still, it is important to note that the main focus of the study by Chamorro and colleagues [21] concerned how long-term immersion modulates the natural pattern of anaphora resolution. Therefore, the only condition that would be informative when compared with the current experiment is the monolingual group. As such, all comparisons between our results and the study by Chamorro and colleagues [21] always refer to the monolingual condition. Chamorro and colleagues [21] observed differences in online interpretation of null and overt pronouns in three eye-tracking measures: *first pass time*, *go-past time*, and *total time*. In the current experiment, we analyzed the same eye-tracking measures. This allowed us to interpret the processes engaged in sentence interpretation, ranging from early (captured by the *first-pass time*) to cumulative (captured by *total time*) measures, which should be sensitive to different stages of syntactic processing.

Based on Experiment 1 and the previous literature, we expected that reading times indexed by *first-pass time*, *go-past time*, and *total time* would be longer for the VPs of sentences in which null pronouns are forced to refer to object antecedents and overt pronouns are forced to refer to subject antecedents as compared to the VPs of sentences in which null pronouns refer to subject antecedents and overt pronouns refer to object antecedents. However, these predictions are not entirely in line with the results of previous studies which showed that null and overt pronouns exhibit different sensitivity to a violation of their referential preferences. Based on the results of a previous eye-tracking experiment by Chamorro and colleagues [21], we expected that overt pronouns would be sensitive to a violation of their resolution preference, while null pronouns would be more flexible and would more easily accommodate the mismatching antecedent. On the other hand, based on the results of the study by Filiaci and collaborators [5], the opposite effect could be expected: in their experiment a processing penalty was observed in the case of violation of the referential preference of a null pronoun. At the same time, Filiaci and colleagues showed cross-linguistic differences in the sensitivity of an overt pronoun to a violation of its resolution preference. In contrast, no such differences were observed for a violation of null pronoun preferences.

To sum up, based on previous experiments, it seems plausible that a processing penalty will be observed when a pronoun is forced to refer to a less-preferred antecedent. However, the discrepancy between the results of previous studies suggests that, due to nuances in the experimental design or cross-linguistic differences, null and overt pronouns may exhibit different sensitivity to a violation of their preferences. As revealed by analysis of the naturalness of the sentences used in Experiment 1, Polish speakers tend to interpret sentences containing null pronouns as more natural than those containing overt pronouns. As such, if Polish pronouns differ in their sensitivity to a violation of a preferred antecedent match, it is plausible to expect that forcing a less-preferred antecedent match in the case of generally more-natural null pronouns will incur smaller processing costs for a reader than violation of the preferred antecedent match in the case of less-natural overt pronouns.

Additionally, similarly to Experiment 1, the participants were asked to rate sentence naturalness after each sentence. We expected the results of the naturalness rating to reflect the online anaphora resolution bias. Specifically, we expected that null pronoun sentences would obtain higher naturalness scores than overt pronoun sentences, and sentences following the preference for anaphora resolution (i.e., a null pronoun referring to the most-prominent antecedent and an overt pronoun referring to a less-prominent antecedent) would obtain higher scores than those violating it. The data and the script necessary to reproduce the statistical analysis for Experiment 2 are freely available online: https://osf.io/dynwp/.

## Materials and methods

**Participants.** Thirty-six native Polish speakers took part in the experiment. Before data analysis, 3 participants were excluded because their second-language proficiency and use were higher than allowed by our criteria. The reported analysis was carried out using the data from 33 participants (31 females, 2 males, age M = 22.15, SD = 2.95). Unlike in Experiment 1, we only recruited participants who did not declare high proficiency or frequent use of a foreign language. While high proficiency or frequent use of a foreign language should not cause any problems in an experiment measuring the preference for anaphora resolution in which respondents have unlimited time to respond, it has been shown that near-native proficiency and immersion in a second language can influence online measures of the anaphora resolution mechanism [2, 21]. Therefore, only participants who reported proficiency lower than B2 and occasional use of their second language (less than 1–2 days per week) took part in the experiment. Apart from the initial pre-screening of language proficiency and use, during the experimental session we assessed participants' proficiency in 5 language skills in the first language (L1: Polish) and the second language (L2: English) with a questionnaire (see Table 2). Participants were recruited from among students of Jagiellonian University and received course credits or monetary remuneration. All participants provided written consent for participation in the experiment and were informed that they could withdraw their consent at any point of the procedure. The experiment met the requirements of the Ethics Committee of the Institute of Psychology of Jagiellonian University concerning experimental studies with human subjects.

**Stimuli.** Following Chamorro and colleagues [21], we created 32 experimental sentences in Polish. Each sentence consisted of the main clause, which contained a subject and an object antecedent of the same grammatical gender but a different grammatical number, and the subordinate clause. The subordinate clause was introduced by a temporal conjunction (*kiedy*–"when"), and it contained a verb conjugated in the third-person singular preceded by an overt or null pronoun. The antecedents were always common nouns depicting people (such as professions, social roles, etc.); proper names were never used. We manipulated the grammatical number of the antecedents: in half of the sentences, the subject was singular and the object was plural; in the other half, the subject was plural and the object was singular. The pronoun and verb of the subordinate clause were always singular, which allowed only one interpretation of the subordinate clause as referring either to the subject or the object of the main clause, depending on which of them was singular in a given sentence. The verbs in the subordinate clause were matched in terms of length and frequency. The length of the main clause, and the frequency of nouns corresponding to the subject and object antecedents, were also matched between sentences. Note that if a sentence included a null pronoun (which is equivalent to omitting the pronoun), no empty space or other replacements were provided in place of the pronoun in the sentence. Each sentence followed the same structure:

subject / verb1 / object / time conjunction ("when") / pronoun (null* or overt) / verb2 / rest1 / rest2

**Table 2. Proficiency in the selected skills in L1 (Polish) and L2 (English) reported by the participants in a self-rated questionnaire.**

| skill | understanding spoken language | understanding written language | fluency of speaking | accent | writing |
|---|---|---|---|---|---|
| **L1 (Polish)** | 9.79 *(0.56)* | 9.69 *(0.97)* | 9.48 *(1.45)* | 9.34 *(1.47)* | 9.38 *(1.37)* |
| **L2 (English)** | 5.79 *(1.97)* | 6.62 *(1.61)* | 4.79 *(1.86)* | 4.62 *(1.80)* | 5.45 *(1.50)* |

Participants were asked to assess their proficiency in respect to each of the 5 language skills on a scale from 0 to 10. Values in brackets correspond to standard deviation.

By manipulating the *Pronoun* and the *Antecedent* match, four different versions of each sentence were created (6–9; conditions following the less-preferred pronoun-antecedent match are marked with?). Note that in Polish the nominative case is always used for a noun in the grammatical position of a subject. In our experiment, the object antecedent always uses the instrumental case; however, this is not a rule as the case of the object of a sentence depends on the verb directly preceding it.

Condition 1: **? overt pronoun / subject match:**

(6) Nauczyciel   porozmawia-ł z   uczniami,   kiedy on wrócił   do szkoły   jesienią.

The teacher    talked   with the students    when he came back   to school   in the fall.

Teacher.NOM.SG talk-1SG.M.PST   with students.INS.PL when   he come back.1SG.M.PST   to school.GEN.SG fall-in-the.

Condition 2: **overt pronoun / object match:**

(7) Nauczyciele   porozmawia-li z   uczniem,   kiedy on wrócił   do szkoły   jesienią.

Teachers    talked   with the student   when he came back   to school   in the fall.

Teachers.NOM.PL    talk-1PL.M.PST   with student.INS.SG   when   he came back.1SG.M.PST to school.GEN.SG fall-in-the.

Condition 3: **null pronoun / subject match:**

(8) Nauczyciel porozmawia-ł z   uczniami,   kiedy   Ø   wrócił   do szkoły   jesienią.

The teacher    talked   with the students when he   came back   to school   in the fall.

Teacher.NOM.SG talk-1SG.M.PST with students.INS.PL when Ø came back.1SG.M.PST to school. GEN.SG fall-in-the.

Condition 4: **? null pronoun / object match:**

(9) Nauczyciele porozmawia-li z   uczniem   kiedy   Ø   wrócił do szkoły   jesienią.

Teachers    talked   with the student   when   Ø   came back   to school   in the fall.

Teachers.NOM.PL    talk-1PL.M.PST   with student.INS.SG when   Ø   came back.1SG.M.PST   to school.GEN.SG    fall-in-the.

Subsequently, four different stimuli lists of 32 sentences were created. Each list contained eight sentences corresponding to each of the four conditions (i.e., eight sentences in Condition 1, 2, 3, and 4). A complete list of experimental sentences is provided in S1 Appendix. Additionally, 64 filler sentences were created and added to the stimuli lists. They contained different grammatical structures, inanimate referents, proper names, and plural pronouns. Each of the lists contained the same 64 filler sentences. In total, each stimuli list contained 96 sentences (32 experimental + 64 filler sentences).

**Procedure.**   The experiment was run using an Eyelink 1000 Desktop Mount eye-tracking system by SR Research. The stimuli were displayed on a 24" screen (BenQXL2411) with screen resolution set to 1920x1080. The screen was situated approximately 70 cm from participants. The sampling rate was set at 1000 Hz. Eye movements were recorded from the right eye only (except for one participant, for whom the calibration for the right eye failed, therefore the movements of the left eye were recorded instead). Before the experiment started, a calibration process was carried out until it was successful. The calibration was repeated during the main task (between trials) whenever necessary. Each trial started with a drift-correction point displayed in the center of the screen. Subsequently, a fixation point was displayed for 500ms on the left side of the screen in the place corresponding to the beginning of the sentence that appeared after it. Participants were instructed to read each sentence carefully and press a mouse button when finished. Subsequently, they were asked to rate the sentence they had just read on a 1–5 naturalness scale by clicking on the corresponding number displayed on the screen. Before the experiment proper, participants completed three training trials. The entire procedure, including calibration and training, took approximately 45 minutes.

**Data analysis.**   Prior to the analysis of eye-tracking measures, we conducted a standard 2-step cleaning procedure using DataViewer software (SR Research). First, we merged all the

fixations shorter than 80ms with the adjacent fixations if the distance between them was shorter than 0.5 angular degrees, which constituted 0.26% of the total fixation number. Subsequently, we deleted all the fixations shorter than 80ms or longer than 1200ms, as well as fixations that fell outside any of the regions of interest; this constituted a further 0.54% of the data.

We conducted the analyses on two different types of data: eye-tracking data and behavioral data (naturalness rating of each sentence). For the eye-tracking analysis, following the experiment by Chamorro and colleagues [21], we chose three measures: *first-pass time* (summed duration of all the fixations in a particular region from the first time the eye enters the region until it leaves the region in either left or right direction); *go-past time* (the sum of all of the fixations after the first entry into an area of interest, including regressions to previous regions, as well as returns to the area of interest in question until the eye enters any subsequent region); and *total time* (sum of all the fixations in a particular region during the whole trial). *First-pass time* is presumed to be sensitive to relatively early processes in sentence comprehension [42, 43]. *Go-past* time is a hybrid measure sensitive to both early and late processes in sentence comprehension. Therefore, these measures should be informative about the processing load related to the integration of a given word and the reader's predictions [44], which are based on native Polish speakers' offline preference for anaphora resolution. *Total time* is a measure that is sensitive to the overall difficulty of processing the relevant word; it is related not only to the automatically constructed predictions but also to conscious reflection on the naturalness and adequacy of a given syntactic structure.

For the eye-tracking data, we performed separate analyses for null and overt pronoun sentences. We found comparing the overt and null pronoun conditions in one analysis problematic due to differences in the size of the critical area of interest. Namely, the verb phrase of the subordinate clause contained only a verb in sentences with a *null* pronoun or a pronoun and a verb in sentences with an *overt* pronoun. As such, the reading times in the overt pronoun condition would always be longer than in the null pronoun condition due to the additional word. Therefore, we decided to analyze overt and null pronoun sentences separately. Even though this did not allow us to compare the differences in overt and null pronoun sentences directly, we were still able to focus on the most important and informative aspects of the present experiment: the influence of the pronoun-antecedent match on online sentence interpretation. We performed separate analyses for both null and overt pronoun conditions for two areas of interest: **verb2** (or pronoun + verb in case of overt pronoun sentences) and **rest1**. The analyses of eye-tracking data were performed using linear mixed-effect models, as implemented in the lme4 package (version: 1.1.26; [34]). Each model was fitted using one within-participant factor, *Antecedent*, with two levels, *subject* and *object*. As *Antecedent* was a categorical predictor, it was dummy coded prior to running the analysis. The effect sizes (Cohen's *d* statistic) were calculated using the lme.dscore function from the EMAtools package (version: 0.1.3; [37]). Similar to Experiment 1, naturalness data were analyzed using a cumulative ordinal model with flexible thresholds created in the Stan computational framework (http://mc-stan.org/) implemented in the brms package (version 2.15.0; [35]). The model included a main effect of *Pronoun* (null or overt), a main effect of *Antecedent* (subject or object), as well as an interaction between the main effects. Random intercept and slopes for *Pronoun* by participants and items were included in the model.

## Results

**Eye-tracking task.**   *Overt pronoun sentences*: *Subject and object antecedent interpretation*. The present section presents the analyses conducted for the sentences with an overt pronoun in which the area of interest contained the pronoun and the verb of the subordinate clause. We

## Overt pronoun sentences

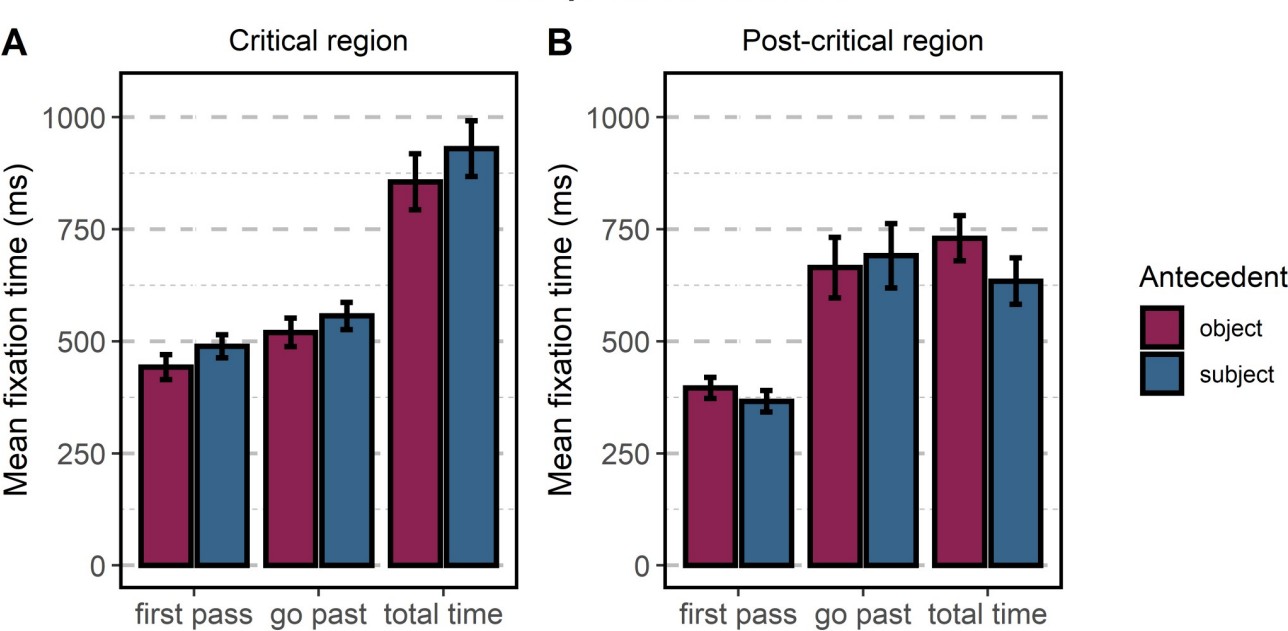

**Fig 4. Mean fixation times for sentences containing an overt pronoun.** (A) The eye-tracking measures for the critical area of interest (pronoun + verb). (B) The eye-tracking measures for the post-critical area of interest (*rest1*). Error bars represent standard errors of the model's predictions.

fitted a separate linear mixed-effects model for each eye-tracking measure with one within-participant factor: *Antecedent*. We found a significant effect of *Antecedent* for *first-pass time* ($t = 2.52$) and *total time* ($t = 2.109$). In both cases, the reading times were longer for sentences with a subject antecedent. We did not find a significant effect of *Antecedent* for *go-past time* ($t = 1.288$). The results for the three measures are presented in Fig 4A and Table 3.

Furthermore, we conducted analyses in the area of interest immediately following the critical region: the post-critical region (*rest1*). In each sentence, this area of interest always

**Table 3. Linear mixed-effect models' estimates for fixed and random effects for the *first-pass*, *go-past*, and *total time* analyses for the critical area of interest in sentences containing overt pronouns.**

| | Fixed effects | | | | | | | | | | | |
| | First pass | | | | Go past | | | | Total time | | | |
| | *Estimate* | *SE* | *t* | *d* | *Estimate* | *SE* | *t* | *d* | *Estimate* | *SE* | *t* | *d* |
| *Intercept* | 466.01 | 25.21 | 18.49 | | 538.49 | 27.69 | 19.45 | | 893.13 | 59.85 | 14.92 | |
| *Antecedent* | 46.54 | 18.43 | 2.53 | 1.21 | 36.73 | 28.51 | 1.29 | 0.47 | 74.08 | 35.12 | 2.11 | 0,19 |
| | Random effects | | | | | | | | | | | |
| | *Variance* | *SD* | *Correlation* | | *Variance* | *SD* | *Correlation* | | *Variance* | *SD* | *Correlation* | |
| by subject | | | | | | | | | | | | |
| *Intercept* | 17237 | 131.29 | | | 18022 | 134.20 | | | 93121 | 305.20 | | |
| *Antecedent* | 1603 | 40.03 | -0.28 | | 11178 | 105.70 | -0.11 | | - | - | - | |
| by item | | | | | | | | | | | | |
| *Intercept* | 1561 | 39.51 | | | 3136 | 56.00 | | | 93121 | 305.20 | - | |
| *Antecedent* | 1287 | 35.87 | -0.09 | | - | - | - | | - | - | - | |

**Table 4. Linear mixed-effect models' estimates for fixed and random effects for *first-pass*, *go-past*, and *total time* analyses for the post-critical area of interest in sentences containing overt pronouns.**

| | Fixed effects | | | | | | | | | | | |
|---|---|---|---|---|---|---|---|---|---|---|---|---|
| | First pass | | | | Go past | | | | Total time | | | |
| | *Estimate* | *SE* | *t* | *d* | *Estimate* | *SE* | *t* | *d* | *Estimate* | *SE* | *t* | *d* |
| *Intercept* | 381.22 | 22.54 | 16.91 | | 677.94 | 61.67 | 10.99 | | 682.33 | 47.33 | 14.42 | |
| *Antecedent* | -29.85 | 15.25 | -1.96 | 1.12 | 26.20 | 64.59 | 0.41 | 0.31 | -95.95 | 38.81 | -2.47 | 0.22 |
| | Random effects | | | | | | | | | | | |
| | *Variance* | *SD* | *Correlation* | | *Variance* | *SD* | *Correlation* | | *Variance* | *SD* | *Correlation* | |
| by subject | | | | | | | | | | | | |
| *Intercept* | 10616.40 | 103.04 | | | 48883 | 221.10 | | | 50192 | 224.04 | | |
| *Antecedent* | - | - | - | | - | - | - | | 5991 | 77.40 | 0.05 | |
| by item | | | | | | | | | | | | |
| *Intercept* | 4948.40 | 70.35 | | | 52547 | 229.20 | | | 18058 | 134.38 | | |
| *Antecedent* | 543.80 | 23.32 | 0.06 | | 34714 | 34714 | 186.30 | | 8917 | 94.43 | 0.11 | |

contained two words: a preposition and a noun or adjective. For each eye-tracking measure, we fitted a separate linear mixed-effects model with one within-participant factor: *Antecedent*. We found no significant effects of *first-pass time* ($t$ = -1.96) or *go-past time* ($t$ = 0.41); however, a significant effect of *Antecedent* was observed for *total time* ($t$ = -2.47). Interestingly, it was in the opposite direction compared to the effects observed for the critical area of interest: reading times were longer for sentences referring to an object antecedent than for those referring to a subject antecedent. The results for the three measures are presented in Fig 4B and Table 4.

*Null pronoun sentences*: *Subject and object antecedent interpretation*. This section presents the analyses conducted for sentences with a null pronoun in which the area of interest contained only the verb of the subordinate clause. We fitted a separate linear mixed-effects model for each eye-tracking measure with one within-participant factor: *Antecedent*. We did not find a significant effect of *Antecedent* for *first-pass time* ($t$ = 1.28), *go-past time* ($t$ = 0.13), or *total time* ($t$ = -0.54). The results for the three measures are presented in Fig 5A and Table 5.

We also analyzed the area of interest immediately following the critical region (*rest1*). For each eye-tracking measure, we fitted a separate mixed-effects model with one within-participant factor: *Antecedent*. Similar to the analysis of the critical region, we did not find a significant effect of Antecedent for *first-pass time* ($t$ = -1.60) or *go-past time* ($t$ = -1.04). However, we found a significant effect of *total time* ($t$ = -2.22). Similar to the effects observed in the post-critical area of interest in sentences containing overt pronouns, reading times for *total time* were longer for sentences referring to the object antecedent. The results for the three measures are presented in Fig 5B and Table 6.

**Naturalness judgment task.** For the analysis of naturalness ratings, we fitted a cumulative ordinal model with two within-participant factors, *Pronoun* and *Antecedent*, and their interaction. Random intercept and slopes by participant and by item were included in the model. Unlike in the eye-tracking analysis, we were able to include both factors in one analysis as the uneven number of words in the subordinate clause in sentences containing overt and null pronouns was not problematic for the behavioral data. The means for each condition are presented in Table 7. We found a strong effect of *Pronoun* ($\beta$ = -0.40; 95% CrI [-0.69–0.10]): participants rated sentences containing null pronouns as more natural than sentences containing overt pronouns. We also found evidence for the interaction of *Pronoun* and *Antecedent* effects ($\beta$ = -1.12, 95% CrI [-1.39–0.84]). Direct comparisons between the two *Antecedent* conditions (i.e., subject- and object-matching sentences) revealed that there was no evidence for

## Null pronoun sentences

**Fig 5. Mean fixation times for sentences containing a null pronoun.** (A) The eye-tracking measures for the critical area of interest (Ø + verb). (B) The eye-tracking measures for the post-critical area of interest (*rest1*). Error bars represent the standard errors of the model's predictions.

differences in naturalness scores in sentences containing a null pronoun (β = 0.15, 95% HPD [-0.10 0.42]). Still, there was a strong effect in sentences containing overt pronouns (β = 1.27, 95% HPD [0.99 1.52]): participants rated subject-matching sentences as less natural than object-matching sentences (see Table 7 and Fig 6).

**Experiment 2 –Summary.** The results of Experiment 2 revealed that in Polish speakers the violation of a preferred pronoun-antecedent match did not result in increased difficulty in processing sentences containing null pronouns. However, in the case of sentences containing

**Table 5. Linear mixed-effect models' estimates for fixed and random effects for the *first-pass time*, *go-past time*, and *total time* analyses for the critical area of interest in sentences containing null pronouns.**

| | Fixed effects | | | | | | | | | | | |
|---|---|---|---|---|---|---|---|---|---|---|---|---|
| | **First pass** | | | | **Go past** | | | | **Total time** | | | |
| | *Estimate* | *SE* | *t* | *d* | *Estimate* | *SE* | *t* | *d* | *Estimate* | *SE* | *t* | *d* |
| *Intercept* | 325.75 | 19.55 | 16.66 | | 371.786 | 23.137 | 16.069 | | 695.71 | 53.1 | 13.103 | |
| *Antecedent* | 19.67 | 15.36 | 1.28 | 0.99 | 2.593 | 19.394 | 0.134 | 0.47 | -16.82 | 31.26 | -0.538 | 0.19 |
| | Random effects | | | | | | | | | | | |
| | *Variance* | *SD* | *Correlation* | | *Variance* | *SD* | *Correlation* | | *Variance* | *SD* | *Correlation* | |
| by subject | | | | | | | | | | | | |
| *Intercept* | 9807 | 99.03 | | | 12716 | 112.76 | | | 71817 | 268 | | |
| *Antecedent* | 3322 | 57.63 | 0.48 | | 3424 | 58.52 | 0.32 | | - | - | - | |
| by item | | | | | | | | | | | | |
| *Intercept* | 1575 | 39.69 | | | 2526 | 50.25 | | | 12276 | 110.8 | | |
| *Antecedent* | - | - | - | | - | - | - | | - | - | - | |

**Table 6. Linear mixed-effect models' estimates for fixed and random effects for the *first-pass*, *go-past*, and *total time* analyses for the post-critical area of interest in sentences containing null pronouns.**

| | Fixed effects | | | | | | | | | | | |
|---|---|---|---|---|---|---|---|---|---|---|---|---|
| | First pass | | | | Go past | | | | Total time | | | |
| | *Estimate* | *SE* | *t* | *d* | *Estimate* | *SE* | *t* | *d* | *Estimate* | *SE* | *t* | *d* |
| *Intercept* | 388.54 | 23.46 | 16.56 | | 694.22 | 67.22 | 10.33 | | 773.68 | 45.88 | 16.86 | |
| *Antecedent* | -24.49 | 15.31 | -1.60 | 1.11 | -72.18 | 69.52 | -1.04 | 0.47 | -71.88 | 32.39 | -2.22 | 0.19 |
| | Random effects | | | | | | | | | | | |
| | *Variance* | *SD* | *Correlation* | | *Variance* | *SD* | *Correlation* | | *Variance* | *SD* | *Correlation* | |
| by subject | | | | | | | | | | | | |
| *Intercept* | 12364.70 | 111.20 | | | 108049 | 328.70 | | | 55203 | 234.95 | | |
| *Antecedent* | - | - | - | | 54657 | 233.80 | -0.93 | | | 54657 | - | |
| by item | | | | | | | | | | | | |
| *Intercept* | 4703.70 | 68.58 | | | 21340 | 146.10 | | | 9555 | 97.75 | | |
| *Antecedent* | 455.30 | 21.34 | - | | - | - | - | | - | - | - | |

overt pronouns, we observed a substantial increase in the cognitive effort whenever a reader was forced to interpret the overt pronoun as referring to the subject antecedent. The naturalness rating supports the eye-tracking results. It revealed that sentences containing null pronouns were rated as equally natural regardless of the pronoun-antecedent match. In contrast, sentences containing overt pronouns which referred to the subject antecedent were rated as much less natural than those referring to the object antecedent. Our results also revealed an unexpected yet very interesting effect: in the analysis of the post-critical area of interest, the reading times were much longer for sentences referring to the object antecedent, regardless of the pronoun-antecedent match. This suggests that processing pronouns which do not refer to the most prominent subject antecedent always incurs additional cognitive effort.

## General discussion

The aim of the research presented in this paper was to investigate the pattern of pronominal anaphora resolution in Polish. In the first experiment, we established the offline preference for interpreting pronominal anaphora in Polish; in the second experiment, we explored whether a violation of this established preference impedes sentence processing. The results of the first experiment reveal that native Polish speakers prefer to interpret sentences containing a null pronoun as referring to a subject antecedent. Native Polish speakers also prefer to interpret sentences containing overt pronouns as referring to an object antecedent. The second experiment extends these findings by showing how the preference for anaphora resolution influences online sentence interpretation. There are two key findings of the second experiment: First, the analysis of the critical area of interest (the pronoun + the verb of the subordinate clause) indicates that a violation of its antecedent preference does not disrupt the interpretation of a null pronoun because a null pronoun can easily be interpreted as referring to the subject and object

**Table 7. Mean scores and SD for the naturalness judgements of the experimental stimuli.**

| Condition | Naturalness rating |
|---|---|
| ***Null*** pronoun–***Subject*** antecedent | 3.77 *(1.13)* |
| ***Null*** pronoun–***Object*** antecedent | 3.89 *(1.06)* |
| ***Overt*** pronoun–***Subject*** antecedent | 2.47 *(1.22)* |
| ***Overt*** pronoun–***Object*** antecedent | 3.53 *(1.10)* |

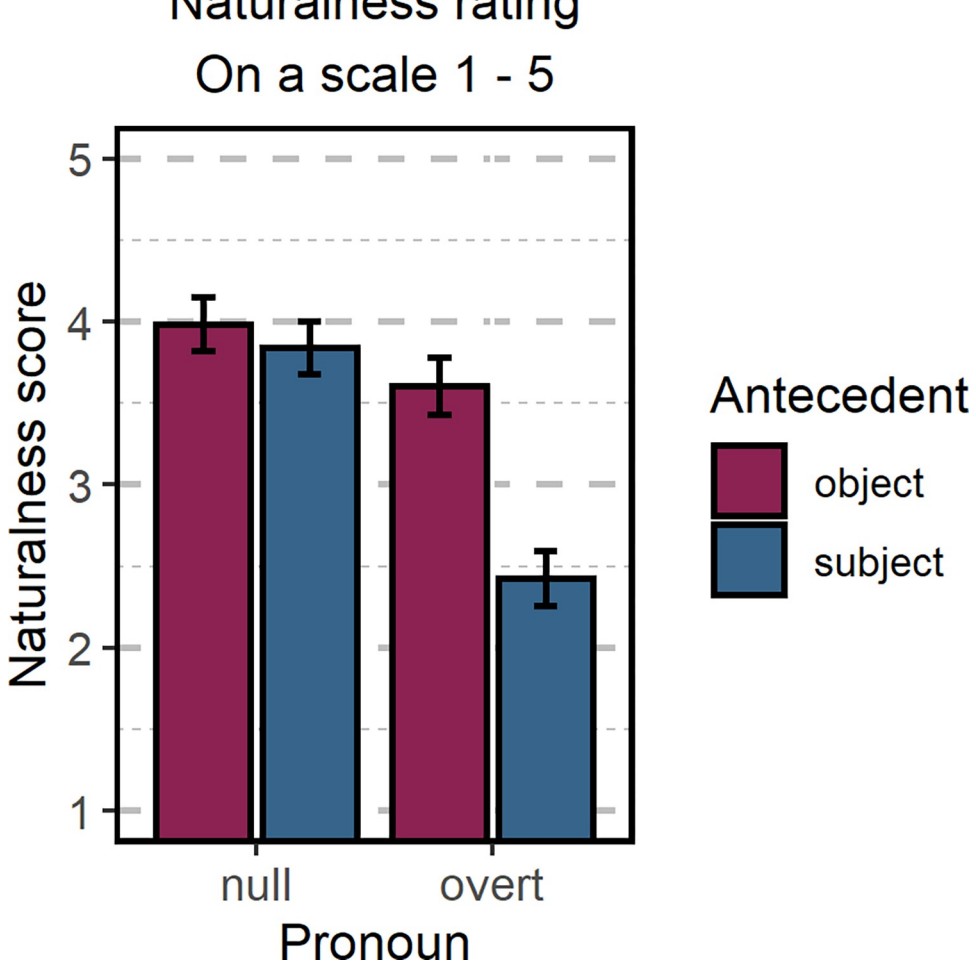

**Fig 6. Naturalness judgment.** The naturalness rating of the sentences used in the eye-tracking experiment. Error bars represent standard errors of the model's predictions.

antecedents. On the other hand, the interpretation of an overt pronoun is much more sensitive to the violation of a pronoun-antecedent match preference: whenever an overt pronoun is forced to refer to a subject antecedent, a substantial increase in processing cost is observed. The second key finding of the second experiment is revealed by the analysis of the post-critical area of interest (the complement of the subordinate clause, following the VP). This analysis shows that, regardless of the pronominal form, sentences in which a pronoun refers to the object antecedent always incur an additional processing cost compared to sentences that refer to the subject antecedent. This suggests that Polish speakers interpret sentences under the working assumption that the most-prominent referent is the first-mentioned subject and topic antecedent. The reference can be shifted towards a different antecedent during the sentence-interpretation process if additional information (like the appearance of an overt pronoun) requires it. However, shifting the predicted reference is always effortful, regardless of the pronoun-antecedent match, which is reflected by increased processing cost in sentences containing overt pronouns.

Additionally, in both Experiments we found that Polish speakers considered sentences containing null pronouns as more natural than sentences containing overt pronouns. Interestingly, our results also show that this effect is modulated by the preference for a pronoun-

antecedent match. In Experiment 1, we found that null pronouns interpreted as referring to subject antecedents were rated as more natural than those interpreted as referring to object antecedents. However, no differences in the naturalness of sentences with null pronouns were observed in Experiment 2. The difference between the two Experiments may be related to differences in the task itself: while Experiment 1 allowed a reader to choose the referent of a pronoun, Experiment 2 forced the pronoun to refer to the only grammatically correct antecedent. In both experiments, we also found that overt pronouns referring to a subject antecedent were rated as much less natural than those referring to an object antecedent (i.e., preferred interpretation of an overt pronoun). This pattern is convergent with our eye-tracking results, which show a higher processing cost for overt pronouns forced to match less-preferred subject antecedents than for null pronouns forced to match less-preferred object antecedents. Based on these findings, it could be argued that sentences that are less natural are actually more difficult to interpret. Consequently, the increased processing cost related to the interpretation of sentences with an overt pronoun that is forced to match a subject antecedent might actually reflect a more general problem related to interpreting very unnatural sentences rather than to shifting the preferred reference of a pronoun. However, this interpretation is challenged by the naturalness judgment results in Experiment 1. Similarly to Experiment 2, the naturalness rating in Experiment 1 shows that Polish speakers consider sentences with overt pronouns that are interpreted as referring to the subject antecedent (i.e., contrary to the general preference) as much less natural than sentences that follow the preference for overt pronoun interpretation. However, in Experiment 1 we did not force the unnatural pronoun-antecedent match: all pronouns were ambiguous and could refer to either of the antecedents. As such, our results show that not only syntactic factors such as subjecthood or first-mention bias drive the perception of the naturalness of sentences. The speaker's interpretation of the sentence, i.e., the pronoun-reference assignment, once it has been established, also does so.

## What drives the anaphora resolution mechanism in Polish?

As revealed by the results of our first experiment, the preference for anaphora resolution in Polish is similar to that observed in other *pro*-drop languages. In Romance languages like Italian or Spanish, speakers usually prefer to interpret sentences containing null pronouns as referring to a subject antecedent, and they interpret sentences containing an overt pronoun as referring to an object antecedent [5, 19, 28]. Our results are also convergent with theoretical accounts according to which the use of a referring expression is determined by the prominence of its referent, which can be derived from one of the following: the referent's grammatical role (the highest SpecIP position; [19]), a preference for the first-mentioned antecedent in a sentence [16, 29]; or the fact that the first-mentioned subjects are more likely than other antecedents to be interpreted as topics of a given utterance [12, 32]. It has been shown that while it is challenging to disentangle these effects, it is very likely that readers use a combination of different cues of prominence when interpreting pronouns (for a comparison of the first-mention account and subjecthood, see [29]). The results of both reported experiments suggest that subjecthood and information structure (i.e., order of mention but also topicality and thematic roles) are dominant constraints for pronoun resolution in Polish. This is a typical observation in sentence-oriented languages which rely on the morpho-syntactic determinants of subject-verb agreement in establishing pronouns' reference. In Experiment 1, we showed that null and overt pronouns are usually bound by antecedents whose prominence can easily be derived from these syntactico-semantic cues. However, the interpretation of ambiguous pronouns also revealed that, in some cases, readers interpreted them contrary to the general resolution preference. Interestingly, even though participants were free to choose whichever antecedent they

preferred, the naturalness scores of the less-preferred antecedent matches (i.e., an overt pronoun referring to a subject antecedent) were much lower than for conditions following a general resolution preference pattern. This result indicates that even though the syntactico-semantic bias for pronoun resolution seems to be the dominant constraint on pronoun interpretation for Polish speakers, it can be overridden by other factors whenever the mental representation of a comprehender's discourse favors an alternative interpretation. Apart from referring to the antecedent's prominence, the preferred pronoun-antecedent match in Polish speakers can also be explained by referring to more pragmatic factors: according to the Accessibility Theory [6], the use of a referring expression depends on the balance between its cost and function. In the case of overt pronouns, the pragmatic function of overt pronouns could be to act as cues which help to identify the appropriate antecedent that is less prominent [45, 46]. As such, whenever a speaker encounters an overt pronoun, it can be interpreted as a signal for a reference shift towards a less-prominent antecedent.

In Experiment 2, we explored the extent to which pronoun resolution preferences can be modulated by the grammatical constraints on the formation of a referential dependency. In the case of Polish, these constraints refer to the grammatical number, person, or gender agreement between the pronoun and its antecedent. Our results indicate that null pronouns can equally easily be interpreted as referring to subject and object antecedents. An important consequence of this is that since null pronouns can be easily and unambiguously interpreted as referring to the antecedent in subject and object positions, any use of an overt pronoun needs to be justified [46]. As previously discussed, the pragmatically driven function of a pronoun is to cue the identification of the appropriate antecedent. In this framework, the appearance of an overt pronoun should signal a shift of reference from the most-prominent antecedent to a less-prominent one. Otherwise, the overt pronoun would provide redundant information and, as such, it should be more difficult to interpret. Our results provide experimental confirmation of this prediction: the observed pattern of online anaphora resolution reveals that interpretation of a sentence containing an overt pronoun that refers to a subject antecedent is much more effortful than in the case of an object antecedent. A similar pattern of results was also reported by a study on online sentence interpretation in Spanish speakers [21]. However, in contrast to Polish and Spanish speakers, Italian speakers show exactly the opposite sensitivity to the violation of a preferred pronoun-antecedent match, with a much bigger processing penalty for a violation of a null pronoun preference than for a violation of an overt pronoun preference [5]. One way of accounting for the differences in the sensitivity of null and overt pronouns to the violation of their preference for interpretation is by referring to approaches which assume that pronoun resolution is driven by multiple constraints, such as the Form-Specific Multiple-Constraint approach [1]. Within these frameworks, our results can be interpreted as indicating that null pronouns are much more sensitive than overt pronouns to a wide range of factors that determine the reference. However, these factors do not necessarily drive the general pronoun resolution preference. Therefore, whenever null pronouns are forced to refer to an antecedent which does not follow the preferred antecedent match, they can easily shift their reference towards a less-preferred referent. On the other hand, overt pronouns do not exhibit similar flexibility which entails a substantial processing penalty whenever the preference for retrieving the object antecedent is violated.

Interestingly, the results of Experiment 2, while replicating the pattern observed by Chamorro and colleagues [21], are at the same time at odds with another study conducted by Filiaci and colleagues. The study by Fialiaci and colleagues showed that a violation of a preferred pronoun-antecedent match incurs a greater processing penalty for null than for overt pronouns [5]. Trying to account for the differences between these two studies, Chamorro and colleagues point out that they may actually be related to the clause order. Similarly to

Chamorro and colleagues, in our experiment we used sentences following the Main-Subordinate clause order. In contrast, Filiaci and colleagues [5] followed the Subordinate-Main order. These differences can affect the information structure as they can shift a reader's expectations, resulting in differences in discourse representations. Furthermore, they can also result in a shift in topicality or a shift of the perceived agentivity of the available antecedents. On the other hand, the discrepancies between our results and those reported by Filiaci and colleagues could be related to cross-linguistic differences in pronoun resolution mechanisms between Polish (our study), Spanish [21], and Italian [5]. Within the multiple-constraint frameworks [1, 24], it can be argued that even though the "hard" morpho-syntactic constraints in different languages are similar (or the same), their anaphora resolution mechanisms might still differ due to differences in sensitivity to the "soft" cues that define the antecedents' prominence (such as grammatical or thematic roles, discourse coherence, etc.).

## The antecedent in an object position–the source of the unexpected processing cost

So far, we have argued that subjecthood, order of mention, and automatic assignment of topicality to the subject antecedent are the dominant constraints on anaphora resolution in Polish. We also showed that in some cases these constraints can be overridden to accommodate the current representation of a discourse better. To our surprise, our data indicate that the processing of sentences referring to the antecedent in the syntactic position of an object incurs an additional processing cost, regardless of the pronoun-antecedent match. Interestingly, this effect was only revealed in the analysis of a post-critical area of interest (the *spillover* effect; [47]). According to the *integration account* [48], an increase in reading times in the post-critical area of interest reflects the cognitive load related to the sentence-level semantic integration process. This explanation is in line with the idea that pronoun resolution is a process that unfolds over time (for a discussion, see [49]): in its early stages, the initial bonding or retrieval is reflected by early fixation measures, but late processes related to integration with the discourse only affect later processing stages. Following the *integration account*, we propose that in our data the spillover effect reflects the cognitive load related to a shift of a *default* preferred reference from the most-prominent subject antecedent to a less-prominent object antecedent. As previously discussed, in our experiment the preferred reference of a given sentence can be derived not only from a preference for a syntactic subject but also from assigning the role of the topic to the first-mentioned antecedent. Under the assumption of the *integration account*, the effects observed in the critical area of interest reflect the ease of processing of the pronoun's *meaning* (i.e., identifying the referent of a verbal phrase). In contrast, the effects in the post-critical region reflect the updating of the syntactic- or discourse-related representation of a sentence. Therefore, even though processing of a less-natural subordinate clause containing an object antecedent might not impose difficulty for lexico-semantic processing, it could generally be more demanding to process on the *syntactic* and *discourse integration* levels. Previous studies on the effect of focus in anaphora resolution reported a similar effect: an increased processing cost associated with a topic shift towards a less-prominent antecedent which has been cued by the information structure of the current discourse [10, 11, 14, 15]. In this context, our results can be interpreted as showing that Polish speakers automatically interpret the most-prominent antecedent (the first available antecedent or the subject antecedent) to be the topic of a sentence; however, as new information cues another antecedent that is coherent with the current discourse, the reference is shifted. This interpretation could be supported by similar findings from an EEG experiment performed by Schumacher and colleagues [50] which showed that co-reference with a less-prominent antecedent results in additional processing

cost reflected by the N400 component, which is responsible for semantic processing and integration of words into a broader discourse. However, they also found that shifting the discourse topic from the antecedent in the initial topical position in a sentence results in an additional late processing cost that is reflected by the Late Positivity component. The authors propose that the N400 effect reflects the pronoun-specific prominence computations necessary for the appropriate resolution of the pronoun, while the Late Positivity is driven by a discourse-internal updating process. To sum up, the effects observed in the post-critical region of interest might indicate that Polish speakers automatically assign the reference (and the role of a topic) of a sentence to the first-mentioned subject antecedent, i.e., the pre-verbal subject of the main clause. While a range of different factors can modulate the interpretation of the pronoun itself, the syntactic or discourse-related representation of a sentence needs to be updated whenever the content of the sentence requires the reference to be shifted towards a different antecedent.

## Conclusions

The current study provides the first experimental evidence on anaphora resolution preferences in Polish. We have shown that the general bias for anaphora resolution is consistent with theories according to which it is the prominence of an antecedent that governs anaphora resolution [1, 6, 19, 46]. Our results indicate that in Polish, syntactico-semantic cues, such as subjecthood and information structure (i.e., order of mention and thematic roles), are the dominant constraints of the prominence of an antecedent; however, they can be overridden by additional cues whenever this is justified by the coherence of discourse representation. Interestingly, the results of the eye-tracking experiment revealed that null pronouns in Polish are able to flexibly and effortlessly shift the reference towards a less-preferred antecedent, while the referential bias for overt pronouns seems to be much stronger as its violation is related to a significant increase in processing costs. Moreover, the analysis of the post-critical region revealed that, regardless of the anaphora resolution process, an additional cognitive effort occurs whenever the reference needs to be shifted towards a less-prominent non-topical antecedent. The fact that this effect is consistent across sentences that contain both null and overt pronouns suggests that it might reflect a different process than the effects observed in the critical area of interest. Altogether, our results suggest that anaphora resolution is an incremental multi-stage process; however, further research is needed to explore the mechanism responsible for the shift of reference in order to better understand the source of the extra cognitive effort it entails.

## Supporting information

**S1 Appendix. A list of experimental sentences used in Experiment 2.**
(DOCX)

## Acknowledgments

The authors would like to thank all members of our Psychology of Language and Bilingualism Laboratory, LangUsta, who contributed to the research project by discussing the task design and collecting and coding the data. We are also grateful to Michael Timberlake for proofreading and to all participants who took part in the experiments. Special thanks to Michał Remiszewski for coordinating the data collection process and to Marta Ruda as well as the three anonymous Reviewers for their helpful comments and insights.

## Author Contributions

**Conceptualization:** Agata Wolna, Joanna Durlik, Zofia Wodniecka.

**Data curation:** Agata Wolna.

**Formal analysis:** Agata Wolna.

**Funding acquisition:** Zofia Wodniecka.

**Investigation:** Agata Wolna, Joanna Durlik.

**Methodology:** Agata Wolna, Joanna Durlik, Zofia Wodniecka.

**Project administration:** Agata Wolna, Zofia Wodniecka.

**Resources:** Joanna Durlik, Zofia Wodniecka.

**Software:** Agata Wolna.

**Supervision:** Zofia Wodniecka.

**Validation:** Agata Wolna.

**Visualization:** Agata Wolna, Joanna Durlik.

**Writing – original draft:** Agata Wolna, Joanna Durlik, Zofia Wodniecka.

**Writing – review & editing:** Agata Wolna, Joanna Durlik, Zofia Wodniecka.

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
