## [Decision Letter · Decision Letter 0]

3 Jun 2021

PONE-D-21-12740

Pronominal anaphora resolution in Polish: investigating online sentence interpretation using eye-tracking.

PLOS ONE

Dear Dr. Wolna,

Thank you for submitting your manuscript to PLOS ONE. After careful consideration, we feel that it has merit but does not fully meet PLOS ONE’s publication criteria as it currently stands. Therefore, we invite you to submit a revised version of the manuscript that addresses the points raised during the review process.

We look forward to receiving your revised manuscript.

Kind regards,

Claudia Felser, Ph.D

Academic Editor

PLOS ONE

Journal Requirements:

2. While the reviewers find your study interesting and note several positive aspects of it, they also raise a number of non-trivial concerns which I am asking you to address in a revised version of your manuscript. Major comments relate to the statistical methods used (which are out-dated and deemed inapppropriate by two reviewers), the coverage and depth of the literature review and discussion, the distinction made between syntax and pragmatics/discourse, and the consideration of alternative models or explanations for the results. Please also respond to the reviewers' more minor comments.

3. Thank you for including your ethics statement:  "The experiment received a written approval of the Ethics Committee of the Institute of Psychology of Jagiellonian University concerning experimental studies with human subjects.".   

Please provide additional details regarding participant consent. In the ethics statement in the Methods and online submission information, please ensure that you have specified what type you obtained (for instance, written or verbal, and if verbal, how it was documented and witnessed). If your study included minors, state whether you obtained consent from parents or guardians. If the need for consent was waived by the ethics committee, please include this information.

4. We note you have included a table to which you do not refer in the text of your manuscript. Please ensure that you refer to Table 4 in your text; if accepted, production will need this reference to link the reader to the Table.

Reviewers' comments:

Reviewer's Responses to Questions

**Comments to the Author**

1. Is the manuscript technically sound, and do the data support the conclusions?

Reviewer #1: Partly

Reviewer #2: Partly

Reviewer #3: Yes

2. Has the statistical analysis been performed appropriately and rigorously? 

Reviewer #1: No

Reviewer #2: No

Reviewer #3: Yes

3. Have the authors made all data underlying the findings in their manuscript fully available?

Reviewer #1: Yes

Reviewer #2: Yes

Reviewer #3: Yes

4. Is the manuscript presented in an intelligible fashion and written in standard English?

Reviewer #1: Yes

Reviewer #2: Yes

Reviewer #3: Yes

5. Review Comments to the Author

Reviewer #1: This paper reports an offline and an online experiment on the interpretation of anaphoric pronouns in Polish. The authors are argued that the results show that interpretation of a null pronoun would be more sensitive to discourse-level cues than syntactic cues, while resolution of overt pronouns would rely strongly on syntax-based cues. While potentially interesting, the paper presents several weaknesses that would require some important improvements before it can be considered for publication in PLOS One.

Detailed comments:

- I do not understand the term “natural preference” (or “natural pattern”) for anaphora resolution used by the authors throughout the manuscript. In most cases, the adjective “natural” can be simply removed. In some other cases it can be replaced by “offline preference” or “baseline preference”.

- p. 8-9, l186-192 and throughout the manuscript, the authors present the null pronoun preference for subject antecedents and the overt pronoun preference for object antecedents as syntactic preferences. This is the hypothesis of the Position of Antecedent Strategy (PAS) but it is not the only one interpretation of the division of labour between null and overt pronouns observed in null-subject languages. In sentences such as those used in the study (e.g., The mother waved to the daughter when Ø (she) was crossing the street.), the preference of the null pronoun for the more salient antecedent (here, the mother) can be interpreted as a preference for the subject but also as a preference for the first mentioned referent, for the topic, for the agent or all at once. The division of labour between null and overt subject pronouns does not demonstrate that only syntactic cues are in play, it may also be driven by semantic or discourse-level cues. The authors should discuss the alternative hypotheses to the PAS and take them into account when interpreting their results.

- p. 8, l165-168, “This approach provides a plausible explanation of the divergent pattern of overt pronoun interpretation that is observed not only between Italian and Spanish (5) but also between sentence-oriented and discourse-oriented languages, which use different types of cues to assign the reference to an anaphoric expressions (for discussion, see: (3))”. Referring to the discussion of Kwon and Sturt is not enough here. The authors should clearly explain how the Form-Specific Multiple-Constraint Approach can account for the cross-linguistic differences in overt pronoun interpretation.

- Results. For both experiments, the authors performed separate F1 and F2 ANOVA analyses. This is not the most appropriate for psycholinguistic experimental data with two random variables (participant and item). The authors should analysed their data with mixed-effects models which allow for simultaneously modelling crossed participant and item effects in a single analysis (see Baayen & al., 2008; Barr, & al., 2013).

- p. 17, l390. For another experiment which used eye-tracking during reading to explore the online pronoun resolution process, see Roberts et al. (2008).

- The authors rely on the PAS to predict the offline preferences: the null pronoun should be preferentially interpreted as referring to the subject whereas the overt pronoun should be preferentially interpreted as referring to the object, which the offline results confirmed. However, they predict (and observe) different online preferences. Their online predictions are based on previous eye-tracking during reading results (see p. 17, l406-412). The discrepancy between online and offline results should be more discussed both in predictions and results.

- Naturalness judgments. An effect of the pronoun form was observed in the naturalness judgments in Study 1 (p. 15-16, l362-366). Was the factor Antecedent (subject vs object) taken into account in this analysis? If I understand correctly, even if the pronoun is ambiguous, the naturalness judgment was asked after the participants had chosen an antecedent for the pronoun. It would be interesting to compare the naturalness judgements observed in Study 1 with those observed in Study 2. In the paper, there is very little discussion of the results of the naturalness judgement results.

- Interpretation of the results of study 2. “Exploring the online interpretation of unambiguous sentences whose only possible interpretation either was or was not in line with the preference for interpretation of anaphorical sentences allowed us to determine whether these preferences are strictly bound to syntax-based rules or whether they can be modulated by the context of a given sentence.” (p. 28, l667-671). Based on the assumption that a syntax-based preference is more costly to change than a discourse-based preference, the authors conclude “we have shown that overt pronouns are much more sensitive than null pronouns to syntax-based determinants of the prominence of an antecedent. In contrast, null pronouns are more sensitive to discourse-related information as they are able to flexibly and effortlessly shift the reference, even against syntax-based cues.” (p. 31, l746-749). However, this assumption was not directly tested in the study (discourse-level factors were not manipulated and disentangle from syntactic cues). The authors’ rationale needs to be more documented and motivated by experimental and/or theoretical arguments. And the authors should consider other possible interpretations of the findings.

Minor points:

- p. 5, l103. The following order seems more logical: “(i.e. null, overt pronouns, stressed, etc.)”

- The authors are missing for the reference (14).

References

Baayen, R. H., Davidson, D. J., & Bates, D. M. (2008). Mixed-effects modeling with crossed random effects for subjects and items. Journal of Memory and Language, 59(4), 390‑412. https://doi.org/10.1016/j.jml.2007.12.005

Barr, D. J., Levy, R., Scheepers, C., & Tily, H. J. (2013). Random effects structure for confirmatory hypothesis testing : Keep it maximal. Journal of Memory and Language, 68(3), 255‑278.

Roberts, L., Gullberg, M., & Indefrey, P. (2008). On-line pronoun resolution in L2 discourse: L1 influence and general learner effects. Studies in Second Language Acquisition, 30, 333–357. https://doi.org/10.1017/S0272263108080480

Reviewer #2: In this paper the authors report two experiments testing the resolution preferences for null and overt pronouns in Polish. Adopting the Position of Antecedent hypothesis, and based on previous studies of anaphora resolution for null and overt pronouns in Italian and Spanish, the authors predict that null pronouns should refer to the subject of the previous (main) clause, and that overt pronouns should refer to the object of the previous (main) clause. In a pronoun interpretation experiment, these predictions were borne out. Naturalness ratings indicated that sentences containing null pronouns were perceived as more natural than sentences containing overt pronouns. In an eye-tracking experiment, violations of these resolution preferences caused longer first-pass and total times in the verb region for the overt pronouns, but had no discernible effect on the null pronouns. The authors conclude that the resolution preferences are in line with the Position of Antecedent hypothesis, and that null pronouns are more flexible in their interpretation than overt pronouns.

I found this a well organised and well written paper with a clear structure. The experiments appear to have been carried out soundly, although some improvements could be made to the analysis (see below). Although brief, coverage of the previous literature was adequate, but the authors should add discussion of a more recent model for pronoun resolution (see below). The findings are quite modest and are reflected fairly in the discussion, it is good that the authors address a gap in the literature by testing pronoun resolution in Polish.

The major points that should be addressed relate to (i) literature coverage; (ii) data analysis; (iii) the interpretation relating to claims about syntax and/or use of terms relating to syntax.

(i) literature coverage

For a short paper, you covered most of the important theoretical background. But you should also include more recent work on pronoun resolution by Kehler and colleagues (e.g. Kehler et al 2008; Kehler and Rohde 2013) as this represents an important development in models of pronoun resolution.

(ii) data analysis

ANOVA: results (especially from Experiment 1) are very clear and I don’t doubt them, but ANOVA has been replaced by linear mixed-effects models as the standard way to analyse psycholinguistic data of this type. This is because mixed models take better account of the random structure of the data, rather than averaging over items and participants. I think your analyses using ANOVA in experiments 1 and 2 should be replaced in order to conform to current standards (I don’t think that your interpretation will change much). If you are not familiar with this type of analysis, a good place to start is Cunnings (2012): An overview of mixed-effects statistical models for second-language researchers. https://journals.sagepub.com/doi/pdf/10.1177/0267658312443651

Power analysis and effect sizes: please report effect sizes. Also, did you do a power analysis? It is good practice to do this in advance of an experiment. If you did, please report it. If not, please consider it for your next experiments; some journals are requesting this as standard.

Analysis of naturalness data: this type of data should not be analysed using raw scores. Z-scores are more appropriate (see Schütze and Sprouse 2014, Judgment Data, DOI: 10.1017/CBO9781139013734.004). Even better would be some kind of ordinal model analysis (see this recent paper: Veríssimo, Analysis of rating scales: A pervasive problem in bilingualism research and a solution with Bayesian ordinal models, on psyarxiv.com)

Reporting main effects: please state which direction the effects go in

(iii) syntax

I think this may just be a terminology issue, but I was confused by your claims about syntax and syntactic cues versus discourse/pragmatics. For instance, in lines 184-192, I really don’t understand the claim you are making here. Why should violation of syntactic cues lead to processing costs, but discourse violations not induce such costs? Is there any prior evidence to that effect? I would expect that a violation of a discourse-based expectation could induce processing costs just as much as a syntactic violation. But further, I don’t understand the distinction you are making between syntax and discourse/pragmatics. As I see it, pronouns are subject to certain syntactic rules in certain configurations, such as Condition B. But this is not what you test in the current study. You seem to equate syntax with a strong preference for a certain grammatical category. Could you clarify what you want to claim here? This issue recurs throughout the paper. For example, in lines 246-251: Previous studies have shown that… Are you making the claim that null pronouns are syntactically resolved and overt pronouns are influenced by non-syntactic factors? How can this be distinguished from discourse-based prominence? I’m not sure that there is such a strict distinction as you are trying to make. Similarly, lines 387-388: syntax-based determinants. Not sure what this is based on. And lines 698-700: overall I agree with the line of argumentation in the discussion, but I don’t agree that the preferences for the resolution of overt pronouns is necessarily best characterised as syntax-based. You can have more rigid interpretation preferences (see work on German demonstratives as an example) without invoking syntactic rules. In my view, not referring to the most prominent antecedent is not necessarily a rule of syntax. Even if you want to say something like “influenced by syntactic factors”, I don’t think you have the evidence here that this is just about grammatical role.

Line 723 onwards: adjustment of syntactically based assignment: again same issue as above

Other comments

The naturalness data in Exp 1 are interesting. They should be discussed/reflected on in more detail.

Summary of Exp 1 (lines 368-373): it looks like preferences for overt pronoun are actually stronger than for null pronouns. Would be good to test this statistically, and reflect on what this could mean. Also, what you refer to as natural here actually contradicts the naturalness data

Processing slowdown for referring to the object: at least in the example in lines 464-475, the object conditions have a less natural scenario with respect to number: several teachers and one student. A prototypical scenario is one teacher and multiple students. Could this be behind the processing slowdown here, and the lower ratings? Are the other items like this too?

Shifting reference expectations/updating discourse expectations has shown to have effects in ERP (Schumacher et al 2015, Backward and Forward Looking Potential of Anaphors), similar to the object processing cost you discuss at the end of the paper.

Minor issues

Page 3 line 39: denominate —> denote

Lines 64-65: it is not the information about the referent of a pronoun that is morphologically encoded in the verb. It is the gender and number of the predicate’s subject. Similarly, in line 71, it is not clear what you mean here by mismatch between a pronoun and a verb.

Example lines 66-67 needs a gloss

Line 232: referent of a verb —> subject of a verb

Fig 2: labels object-match and subject match are odd because they don’t relate to the experimental set up. Simply subject and object?

Lines 389-390: there are a lot of studies that have used eye-tracking to look at pronoun resolution, should refer to these.

Line 382: higher fixation times: this is a prediction relating to a specific measure (fixation times), but you actually tested first-pass, go-past and total times which are more than simply fixation times.

Line 404: total time is a cumulative, not a late measure (captures both early and late processes).

Around line 415: I think I get what you mean in the predictions for the naturalness ratings, but you need to spell it out more clearly (what you mean by natural here).

Condition examples (lines 464-475): need a gloss

Line 565, figure 4A caption is incorrect - not pronoun + verb (no pronoun because null)

Line 635: syntactic position of an object —> syntactic position of a subject

Reviewer #3: In my opinion, the present study presents several strong points: First, it is a close replication of a previous study (both in terms of tasks and materials); second, it adds a (Slavic) language to the repertoire of languages that have been studied in the pronoun resolution literature; third, the reported results confirm a well-known bias previously attested in other languages.

My concern regarding this article, as I explain just below, has to do with the fact that the literature review and parts of the discussion section are rather superficial (in particular when it comes to the discussion of the factors that play a role in pronoun resolution).

See attached file for further comments.

6. PLOS authors have the option to publish the peer review history of their article (what does this mean?). If published, this will include your full peer review and any attached files.

Reviewer #1: No

Reviewer #2: No

Reviewer #3: No

---

## [Author Response · Author response to Decision Letter 0]

12 Aug 2021

Response to the Editor’s comments:

We made sure that our manuscript meets PLOS ONE’s style requirements and that the uploaded files are named according to the provided guidelines.

2. While the reviewers find your study interesting and note several positive aspects of it, they also raise a number of non-trivial concerns which I am asking you to address in a revised version of your manuscript. Major comments relate to the statistical methods used (which are out-dated and deemed inapppropriate by two reviewers), the coverage and depth of the literature review and discussion, the distinction made between syntax and pragmatics/discourse, and the consideration of alternative models or explanations for the results. Please also respond to the reviewers' more minor comments.

We have carefully considered the Reviewers’ comments and we introduced the changes to our manuscript according to their suggestions. 

(1) First, we reanalyzed our data using the linear mixed-effects models and, in the case of naturalness ratings, the Bayesian ordinal models. 

(2) Second, we extended the literature review by including a discussion of an additional model an by focusing on frameworks alternative to the syntax-based Position of Antecedent strategy. Those changes are reflected in the Introduction as well as in the Discussion of our results. 

(3) Third, we reformulated the problematic distinction between syntax and pragmatics/discourse by centering our narration around two alternative, literature-based distinctions of factors that influence the pronoun resolution. The first distinction, proposed by Kaiser and Trueswell (2008) distinguishes between syntactico-semantic and discourse-related factors. The second distinction, proposed by Kehler and Rhode (2013) distinguishes between “hard” and “soft” constraints on pronoun resolution. Both distinctions are introduced and described in detail in the Introduction; they are also referred to in the discussion of the results of the current study

(4) Finally, we addressed all minor concerns of the Reviewers (see the item-by-item responses to all comments below).

3. Thank you for including your ethics statement: "The experiment received a written approval of the Ethics Committee of the Institute of Psychology of Jagiellonian University concerning experimental studies with human subjects.". 

Please provide additional details regarding participant consent. In the ethics statement in the Methods and online submission information, please ensure that you have specified what type you obtained (for instance, written or verbal, and if verbal, how it was documented and witnessed). If your study included minors, state whether you obtained consent from parents or guardians. If the need for consent was waived by the ethics committee, please include this information.

The ethics statement has been updated in the manuscript and it now provides additional information of the form of consent obtained from participants of both reported Studies. The ethics statement has also been updated in the submission system.

4. We note you have included a table to which you do not refer in the text of your manuscript. Please ensure that you refer to Table 4 in your text; if accepted, production will need this reference to link the reader to the Table.

We made sure that all Tables are properly referred to in the text. Following the suggestion of the Reviewers to reanalyse our data with linear mixed-effects models, we included additional Tables that summarize the results of the statistical analyses, therefore the numeration of Tables was updated.

Response to reviewers

Reviewer #1: 

This paper reports an offline and an online experiment on the interpretation of anaphoric pronouns in Polish. The authors are argued that the results show that interpretation of a null pronoun would be more sensitive to discourse-level cues than syntactic cues, while resolution of overt pronouns would rely strongly on syntax-based cues. While potentially interesting, the paper presents several weaknesses that would require some important improvements before it can be considered for publication in PLOS One.

Detailed comments:

- I do not understand the term “natural preference” (or “natural pattern”) for anaphora resolution used by the authors throughout the manuscript. In most cases, the adjective “natural” can be simply removed. In some other cases it can be replaced by “offline preference” or “baseline preference”.

Response 1.1: According to your suggestion, we have changed the term “natural preference” throughout the manuscript by removing the adjective or replacing it with “offline” whenever we wanted to highlight the contrast between the preferences established in Study 1 and the preferences reflected in the online sentence interpretation in Study 2.

- p. 8-9, l186-192 and throughout the manuscript, the authors present the null pronoun preference for subject antecedents and the overt pronoun preference for object antecedents as syntactic preferences. This is the hypothesis of the Position of Antecedent Strategy (PAS) but it is not the only one interpretation of the division of labour between null and overt pronouns observed in null-subject languages. In sentences such as those used in the study (e.g., The mother waved to the daughter when Ø (she) was crossing the street.), the preference of the null pronoun for the more salient antecedent (here, the mother) can be interpreted as a preference for the subject but also as a preference for the first mentioned referent, for the topic, for the agent or all at once. The division of labour between null and overt subject pronouns does not demonstrate that only syntactic cues are in play, it may also be driven by semantic or discourse-level cues. The authors should discuss the alternative hypotheses to the PAS and take them into account when interpreting their results.

Response 1.2: Thank you for your comment. It is indeed the case that in the previous version of the manuscript the predictions and results were discussed in detail only in reference to the PAS framework (and, generally, in reference to theories that propose syntactic preferences as a dominant constraint on pronominal anaphora resolution). Following the Reviewer’s suggestion, to the Introduction (p. 6 lines 111–126) we have added a paragraph describing the different factors that have been shown to influence anaphora resolution (such as grammatical roles, order of mention, agentivity, topicality, focus, information structure and discourse coherence). We have also included this information when discussing previous experimental evidence on anaphora resolution (p. 11, lines 248-252). We now discuss how the alternative accounts to PoA would translate to predictions for Study 1 (p. 16 lines 370-374) and we consider them when discussing the results of Study 1 (p. 39 lines 915–931) and Study 2 (pp. 41-42, lines 972-999). Moreover, following the suggestions of Reviewer #2 regarding the review of previous literature, we discussed a recent theory of pronominal anaphora resolution which was proposed by Kehel and Rhode (2013) as a reconciliation of coherence-driven and centering-driven approaches (pp. 9-10 lines 211-234) and which differentiates between “hard” morpho-syntactic constraints of pronoun reference (such as gender, number, person etc.) and “soft” constraints (such as grammatical or thematic roles, syntactic function, coherence etc.). The revised discussion of our result also reflects this proposal.

- p. 8, l165-168, “This approach provides a plausible explanation of the divergent pattern of overt pronoun interpretation that is observed not only between Italian and Spanish (5) but also between sentence-oriented and discourse-oriented languages, which use different types of cues to assign the reference to an anaphoric expressions (for discussion, see: (3))”. Referring to the discussion of Kwon and Sturt is not enough here. The authors should clearly explain how the Form-Specific Multiple-Constraint Approach can account for the cross-linguistic differences in overt pronoun interpretation.

Response 1.3: A more elaborate explanation of how the Form-Specific Multiple-Constraint Approach could account for the cross-linguistic differences mentioned in the text has been added to the end of the paragraph describing the theory (p. 9 lines 201-210).

- Results. For both experiments, the authors performed separate F1 and F2 ANOVA analyses. This is not the most appropriate for psycholinguistic experimental data with two random variables (participant and item). The authors should analyze their data with mixed-effects models which allow for simultaneously modelling crossed participant and item effects in a single analysis (see Baayen & al., 2008; Barr, & al., 2013).

Response 1.4: Following the Reviewer’s suggestions, we reanalyzed the data using linear mixed-effects models for both experiments and updated the corresponding Results sections. In terms of the significance of the effects we tested, the analyses using LMMs yield the same pattern of results than the previously used ANOVAs. We have added Tables that summarize the results of the LMMs that we used to analyze the data from Experiment 2 to provide a comprehensive overview of the results [Tables 3–6]. The results of the LMMs analyses did not change significantly with respect to the previously reported results of ANOVAs. Following the suggestion of Reviewer #2, who pointed out that this approach provides a much better fit for data based on ordinal scales, we now use Bayesian ordinal models for the analysis of the naturalness of sentences in both Experiments (we provide a short justification for the use of ordinal models on p.19, lines 445-448). 

- p. 17, l390. For another experiment which used eye-tracking during reading to explore the online pronoun resolution process, see Roberts et al. (2008).

Response 1.5: Thank you for this suggestion. The reference to this text has now been added on p. 11 line 255 and p. 22 line 528 (reference n.30).

- The authors rely on the PAS to predict the offline preferences: the null pronoun should be preferentially interpreted as referring to the subject whereas the overt pronoun should be preferentially interpreted as referring to the object, which the offline results confirmed. However, they predict (and observe) different online preferences. Their online predictions are based on previous eye-tracking during reading results (see p. 17, l406-412). The discrepancy between online and offline results should be more discussed both in predictions and results.

Response 1.6: Thank you very much for raising this important point. Indeed, the predictions referring to online preferences were based on the previous experiments without making a clear reference to our offline experiment. Following your suggestion and a similar comment by Reviewer #3 (R3.26), we modified the predictions for Study 2 to better explain our reasoning behind them. First, we adopt the assumption that a violation of the preference for pronoun resolution in Polish that we established in Study 1 would incur additional processing cost. Second, we explain how previous studies that tested pronouns’ sensitivity to a violation of a preferred antecedent match provide contradictory results and suggest that there are cross-linguistic differences in pronouns' sensitivity to violation of referential preferences. Finally, based on what we show in Study 1, we explain that sentences containing null pronouns are interpreted as more natural than overt pronoun sentences, regardless of the antecedent they refer to. As such, we propose that when Polish pronouns indeed exhibit differences in how strong their preferences for resolution are, it can be expected that the processing penalty will be smaller for null pronouns (which are generally more natural) than for overt pronouns. The necessary changes have been introduced to the text (pp. 23–24, lines 546-571).

- Naturalness judgments. An effect of the pronoun form was observed in the naturalness judgments in Study 1 (p. 15-16, l362-366). Was the factor Antecedent (subject vs object) taken into account in this analysis? If I understand correctly, even if the pronoun is ambiguous, the naturalness judgment was asked after the participants had chosen an antecedent for the pronoun.

Response 1.7: Thank you very much for this suggestion. We have updated our analysis of naturalness in Study 1 so that it shows how naturalness is influenced by the Pronoun and the preferred Antecedent. This analysis revealed a similar pattern to the naturalness analysis in Study 2: we found a main effect of pronoun, which shows that sentences containing a null pronoun are interpreted as more natural than sentences containing an overt pronoun (similarly to Study 2). We also found a significant interaction between Pronoun and Antecedent factor: similarly to Study 2, sentences with overt pronouns were rated as more natural when they were interpreted as matching the object than when they were interpreted as matching the subject antecedent. However, unlike in Study 2, we also observed a significant difference between sentences with null pronouns that were interpreted as matching the subject and sentences with null pronouns that were interpreted as matching the object: the former were rated as more natural. Another difference in respect to Study 2 is that in the naturalness analysis in Study 1 we did not observe a significant main effect of Antecedent. However, it is the interaction between the Pronoun and Antecedent that is crucial and most informative. 

The updated analysis of naturalness scores is now included in the Results of Study 1 (p. 20 lines 473–485).

It would be interesting to compare the naturalness judgements observed in Study 1 with those observed in Study 2. In the paper, there is very little discussion of the results of the naturalness judgement results.

Response 1.8: Following the Reviewer’s suggestion, to the Discussion (p. 37–38, l. 865–891) we added a paragraph discussing the naturalness scores of both Studies (as mentioned in the previous comment) and their relation to the eye-tracking results. Additional mentions of naturalness were also included in further parts of the Discussion, where we consider the mechanisms that drive anaphora resolution in Polish (p. 41 lines 965-969).

- Interpretation of the results of study 2. “Exploring the online interpretation of unambiguous sentences whose only possible interpretation either was or was not in line with the preference for interpretation of anaphorical sentences allowed us to determine whether these preferences are strictly bound to syntax-based rules or whether they can be modulated by the context of a given sentence.” (p. 28, l667-671). Based on the assumption that a syntax-based preference is more costly to change than a discourse-based preference, the authors conclude “we have shown that overt pronouns are much more sensitive than null pronouns to syntax-based determinants of the prominence of an antecedent. In contrast, null pronouns are more sensitive to discourse-related information as they are able to flexibly and effortlessly shift the reference, even against syntax-based cues.” (p. 31, l746-749). However, this assumption was not directly tested in the study (discourse-level factors were not manipulated and disentangled from syntactic cues). The authors’ rationale needs to be more documented and motivated by experimental and/or theoretical arguments. And the authors should consider other possible interpretations of the findings.

Response 1.9: Thank you for pointing out this problem to us. First of all, we fully agree that our design does not make it possible to test which factors (syntax-based or discourse-related) affect the interpretation of null and overt pronouns. To avoid possible confusion, we reformulated the discussion so it is centered around (1) the preferences for anaphora resolution in Polish, and (2) whether these preferences can be easily overridden whenever the morpho-syntactic structure of a sentence requires so. In other words, we now discuss our results on a more general level. At the same time, to avoid potential confusion, we decided to drop the distinction between syntax-based and discourse-related factors. Instead, we refer to two distinctions proposed in previous literature: (1) “hard” morpho-syntactic constraints (e.g. number, gender, person etc.) vs. “soft” constraints or heuristics (such as grammatical or thematic roles, syntactic function, coherence etc.), which were introduced and discussed by Kehler and Rhode (2013) (p. 10 lines 226-234); and (2) syntactico-semantic factors (subjecthood and grammatical roles, thematic roles, agentivity) vs. discourse-related factors (topicalization, focus, order of mention), which were introduced by Kaiser and Trueswell (2008) (p. 6 lines 111–126). This new approach to the discussion made it possible to discuss our results within the existing frameworks, extend our predictions and findings beyond the syntax-based constraint of subjecthood, as well as better define the “discourse-related” factors. All mentions of syntax-based cues throughout the text have been deleted or rephrased to be more specific:

In the case of the examples brought up in your comment, we made the following changes:

The fragment: “Exploring the online interpretation of unambiguous sentences whose only possible interpretation either was or was not in line with the preference for interpretation of anaphorical sentences allowed us to determine whether these preferences are strictly bound to syntax-based rules or whether they can be modulated by the context of a given sentence.”

was changed to: “Exploring the online interpretation of unambiguous pronouns whose only possible interpretation either was or was not in line with the preference for interpretation of anaphorical sentences allowed us to determine whether these preferences can be modulated by the context of a given sentence” (p. 40 lines 947-950).

The fragment: “we have shown that overt pronouns are much more sensitive than null pronouns to syntax-based determinants of the prominence of an antecedent. In contrast, null pronouns are more sensitive to discourse-related information as they are able to flexibly and effortlessly shift the reference, even against syntax-based cues.” was replaced by: “Our results indicate that in Polish syntactico-semantic cues such as subjecthood and information structure (i.e., order of mention but also a typical topicality) are the dominant constraints of the prominence of an antecedent; however, they can be overridden by additional cues whenever this is justified by the coherence of discourse representation.” (p. 45 lines 1083-1086).

Minor points:

- p. 5, l103. The following order seems more logical: “(i.e. null, overt pronouns, stressed, etc.)”

Response 1.10: Thank you. The text has been updated according to your suggestions.

- The authors are missing for the reference (14).

Response 1.11: Thank you for spotting this. It has been fixed (in the current version it’s the reference number 28).

References

Baayen, R. H., Davidson, D. J., & Bates, D. M. (2008). Mixed-effects modeling with crossed random effects for subjects and items. Journal of Memory and Language, 59(4), 390 412. https://doi.org/10.1016/j.jml.2007.12.005

Barr, D. J., Levy, R., Scheepers, C., & Tily, H. J. (2013). Random effects structure for confirmatory hypothesis testing : Keep it maximal. Journal of Memory and Language, 68(3), 255 278.

Roberts, L., Gullberg, M., & Indefrey, P. (2008). On-line pronoun resolution in L2 discourse: L1 influence and general learner effects. Studies in Second Language Acquisition, 30, 333–357. https://doi.org/10.1017/S0272263108080480

###################

Reviewer #2: In this paper the authors report two experiments testing the resolution preferences for null and overt pronouns in Polish. Adopting the Position of Antecedent hypothesis, and based on previous studies of anaphora resolution for null and overt pronouns in Italian and Spanish, the authors predict that null pronouns should refer to the subject of the previous (main) clause, and that overt pronouns should refer to the object of the previous (main) clause. In a pronoun interpretation experiment, these predictions were borne out. Naturalness ratings indicated that sentences containing null pronouns were perceived as more natural than sentences containing overt pronouns. In an eye-tracking experiment, violations of these resolution preferences caused longer first-pass and total times in the verb region for the overt pronouns, but had no discernible effect on the null pronouns. The authors conclude that the resolution preferences are in line with the Position of Antecedent hypothesis, and that null pronouns are more flexible in their interpretation than overt pronouns.

I found this a well organised and well written paper with a clear structure. The experiments appear to have been carried out soundly, although some improvements could be made to the analysis (see below). Although brief, coverage of the previous literature was adequate, but the authors should add discussion of a more recent model for pronoun resolution (see below). The findings are quite modest and are reflected fairly in the discussion, it is good that the authors address a gap in the literature by testing pronoun resolution in Polish.

The major points that should be addressed relate to (i) literature coverage; (ii) data analysis; (iii) the interpretation relating to claims about syntax and/or use of terms relating to syntax.

(i) literature coverage

For a short paper, you covered most of the important theoretical background. But you should also include more recent work on pronoun resolution by Kehler and colleagues (e.g. Kehler et al 2008; Kehler and Rohde 2013) as this represents an important development in models of pronoun resolution.

Response 2.1: Thank you for drawing this work to our attention. A sub-section describing the coherence-driven approach in pronoun resolution proposed by Kehler et al. (2008) and Kehler and Rhode (2013) has been added to the Introduction (pp. 9-10, lines 211-234). 

(ii) data analysis

ANOVA: results (especially from Experiment 1) are very clear and I don’t doubt them, but ANOVA has been replaced by linear mixed-effects models as the standard way to analyse psycholinguistic data of this type. This is because mixed models take better account of the random structure of the data, rather than averaging over items and participants. I think your analyses using ANOVA in experiments 1 and 2 should be replaced in order to conform to current standards (I don’t think that your interpretation will change much). If you are not familiar with this type of analysis, a good place to start is Cunnings (2012): An overview of mixed-effects statistical models for second-language researchers. https://journals.sagepub.com/doi/pdf/10.1177/0267658312443651

Response 2.2: We agree that this is a better approach to data analysis; thank you for motivating us to reanalyze our results using LMMs. We have reanalyzed our data using a linear mixed-effects model (in the case of Experiment 1, we used a generalized linear mixed-effect model because our dependent variable was binomial). The Results sections have been updated and additional tables providing summaries of the models have been added to provide a comprehensive overview of the results [Tables 3–6]. The analyses using LMMs yield the same pattern of results than the previously used ANOVAs in terms of the significance of the effects we tested.

Power analysis and effect sizes: please report effect sizes. Also, did you do a power analysis? It is good practice to do this in advance of an experiment. If you did, please report it. If not, please consider it for your next experiments; some journals are requesting this as standard.

Response 2.3: Unfortunately, we did not run a power analysis prior to our experiments. However, our sample size is already bigger than that of the study by Chamorro and colleagues (2016), which our study was based on (our study, n=36; Chamorro et al. (2016), n = 24).

As for the effect sizes, we calculated Cohen's d for each of the effects included in the mixed models (for Experiments 1 and 2). In the case of the naturalness rating analyses, since (following another comment by the Reviewer) we reanalyzed the data using the Bayesian approach, the coefficients of the Bayesian ordinal models that we now report in the text are equivalent to the effect size estimates. As such, we believe no additional estimates are needed.

Analysis of naturalness data: this type of data should not be analysed using raw scores. Z-scores are more appropriate (see Schütze and Sprouse 2014, Judgment Data, DOI: 10.1017/CBO9781139013734.004). Even better would be some kind of ordinal model analysis (see this recent paper: Veríssimo, Analysis of rating scales: A pervasive problem in bilingualism research and a solution with Bayesian ordinal models, on psyarxiv.com)

Response 2.4: We thank the Reviewer for bringing this issue to our attention. Following this comment, we decided to reanalyze the naturalness data for Experiments 1 and 2 using Bayesian ordinal models, similarly to the approach described in the suggested paper. Thank you for providing this great resource. The Results sub-sections that refer to the naturalness judgement analysis (p. 20, lines 469-485 and p. 34, lines 780-797) as well as the figures corresponding to the naturalness analysis of Experiment 1 (Fig 3) Experiment 2 (Fig 7) have been updated according to the new analysis. In the Methods section for Experiment 1, we have also included a Data Analysis subsection (pp.18-19, lines 433-449) to explain the details of our analytical approach; we have updated the Data Analysis section for Experiment 2 (p. 29, lines 699-709).

Reporting main effects: please state which direction the effects go in

Response 2.5: We added information to the text about the directions of all the main effects described in the Results sections.

(iii) syntax

I think this may just be a terminology issue, but I was confused by your claims about syntax and syntactic cues versus discourse/pragmatics. For instance, in lines 184-192, I really don’t understand the claim you are making here. Why should violation of syntactic cues lead to processing costs, but discourse violations not induce such costs? Is there any prior evidence to that effect? I would expect that a violation of a discourse-based expectation could induce processing costs just as much as a syntactic violation. But further, I don’t understand the distinction you are making between syntax and discourse/pragmatics. As I see it, pronouns are subject to certain syntactic rules in certain configurations, such as Condition B. But this is not what you test in the current study. You seem to equate syntax with a strong preference for a certain grammatical category. Could you clarify what you want to claim here? This issue recurs throughout the paper. For example, in lines 246-251: Previous studies have shown that… Are you making the claim that null pronouns are syntactically resolved and overt pronouns are influenced by non-syntactic factors? How can this be distinguished from discourse-based prominence? I’m not sure that there is such a strict distinction as you are trying to make. Similarly, lines 387-388: syntax-based determinants. Not sure what this is based on. And lines 698-700: overall I agree with the line of argumentation in the discussion, but I don’t agree that the preferences for the resolution of overt pronouns is necessarily best characterised as syntax-based. You can have more rigid interpretation preferences (see work on German demonstratives as an example) without invoking syntactic rules. In my view, not referring to the most prominent antecedent is not necessarily a rule of syntax. Even if you want to say something like “influenced by syntactic factors”, I don’t think you have the evidence here that this is just about grammatical role.

Response 2.6: Thank you for directing our attention to this problem. We agree that the distinction between syntax-based or discourse-related factors in pronoun resolution might have been confusing (and, in some cases, probably ungrounded). As this issue was also brought up by other Reviewers, we decided to omit this distinction. We have introduced two important changes in the text to better explain our interpretation of the factors that influence pronoun resolution in Polish: 

1. Throughout the text we refer to two distinctions proposed in previous literature which describe factors influencing pronominal anaphora resolution: (1) “hard” morpho-syntactic constraints (e.g. number, gender, person etc.) vs. “soft” constraints or heuristics (such as grammatical or thematic roles, syntactic function, coherence etc.), which were introduced and discussed by Kehler and Rhode (2013) (p. 10 lines 227-234); (2) syntactico-semantic (subjecthood and grammatical roles, thematic roles, agentivity) vs. discourse-related (topicalization, focus, order of mention), which were introduced by Kaiser and Trueswell (2008) (p. 6 lines 111-126). We have also provided a short summary of previous findings which show that pronoun resolution can be modulated by a number of different factors (p. 6 lines 111–126). This allowed us to discuss our results within the existing frameworks, extend our predictions and findings beyond the syntax-based constraint of subjecthood, and better define the “discourse-related” factors. All mentions of syntax-based cues have been deleted or replaced by more specific formulations throughout the text. 

2. We reformulated some claims so they are centered around (1) the preferences for anaphora resolution in Polish, and (2) whether these preferences can be easily overridden whenever the morpho-syntactic structure of a sentence requires so. In other words, we discuss our results on a more general level without making specific claims about the nature of the constraints on pronominal anaphora resolution. We attempt to discuss our results while taking into account various alternative explanations which have been proposed to influence the pronoun resolution process (p. 38 lines 900-906; p. 39 lines 908-913; p.41 lines 973-984, p. 42, lines 992-1000).

Line 723 onwards: adjustment of syntactically based assignment: again same issue as above

Response 2.7: We believe that the steps described in the response to your previous comments should have resolved this issue.

Other comments

The naturalness data in Exp 1 are interesting. They should be discussed/reflected on in more detail.

Response 2.8: We extended the analysis of naturalness by including the Antecedent factor (the preferred interpretation of an ambiguous sentence provided by a given subject) and its interaction with Pronoun (null vs. overt), as suggested by Reviewer #1. As such, we were able to compare the naturalness results for both Studies. To the Discussion section of our paper (pp. 37-38 lines 865-891), we have added a paragraph with a more in-depth summary, comparison and discussion of the naturalness scores from both Studies.

Summary of Exp 1 (lines 368-373): it looks like preferences for overt pronoun are actually stronger than for null pronouns. Would be good to test this statistically, and reflect on what this could mean.

Response 2.9: Our understanding is that this comment was driven by the difference between the preference for null and overt pronouns to refer to subject or object antecedents (Fig. 2). However, this Figure presents only the raw means but not the results of the statistical model. As the participants’ preference for sentence interpretation (i.e. subject or object) was a dependent variable in our analysis, we can only estimate the main effect of pronoun (i.e. null or overt). As such, our analysis allows us to determine whether Polish speakers interpret sentences with null and overt pronouns as systematically referring to different antecedents, but we cannot compare the strength of these preferences directly.

Also, what you refer to as natural here actually contradicts the naturalness data

Response 2.10: Following a suggestion from Reviewer #1, we decided not to refer to the preferences established in Study 1 as “natural”. We have adjusted the text so that now we discuss “preferences” and not “natural preferences”. We assume this comment was actually caused by imprecise terminology. We acknowledge that there are important differences between the preference for pronoun resolution and the naturalness analysis; in the revised version of our text, we discuss these discrepancies in more detail in the Discussion (pp. 33–39 lines 918-925) and in the context of the predictions for Study 2, which were based on the results of Study 1 (pp. 23–24 lines 566–571).

Processing slowdown for referring to the object: at least in the example in lines 464-475, the object conditions have a less natural scenario with respect to number: several teachers and one student. A prototypical scenario is one teacher and multiple students. Could this be behind the processing slowdown here, and the lower ratings? Are the other items like this too?

Response 2.11 Thank you for pointing out this important issue; it is indeed a very interesting question. Unfortunately, we did not control for the prototypicality of our stimuli; as data collection is now completed, there is nothing that could be changed to account for this issue at the level of the experimental design. However, to assess the possible scope of the problem, we attempted to classify our sentences as prototypical or not post hoc: we found that 7 out of 32 stimuli might be considered as describing a (more or less) prototypical situation. In 4 of these stimuli, there should prototypically be one subject antecedent; in 3 stimuli, there should prototypically be one object antecedent. While we agree that prototypicality could influence naturalness ratings and sentence processing itself, as there are only a few prototypical sentences in our dataset it is impossible to statistically test this and get meaningful results. For the same reason, we think it is reasonable to assume that prototypicality did not confound the results of our experiment to a degree which would impede our inference related to the pronoun-antecedent manipulation. As such, even though we did not (and at this point we are not able to) control for the prototypicality of our stimuli, we believe they still contribute valuable results to the existing literature. A complete list of experimental stimuli can be found in Appendix A (also posted on osf).

Shifting reference expectations/updating discourse expectations has shown to have effects in ERP (Schumacher et al 2015, Backward and Forward Looking Potential of Anaphors), similar to the object processing cost you discuss at the end of the paper.

Response 2.12: Thank you for providing this very useful reference! We added a discussion of the findings of this study to our text (pp. 44-45 lines 1056-1071).

Minor issues

Page 3 line 39: denominate —> denote

Response 2.13: This has been updated in the text.

Lines 64-65: it is not the information about the referent of a pronoun that is morphologically encoded in the verb. It is the gender and number of the predicate’s subject. Similarly, in line 71, it is not clear what you mean here by mismatch between a pronoun and a verb.

Response 2.14: We have provided an additional explanation to account for these problems. In the first case mentioned in your comment, the previous formulation (“[...] information about the referent of a pronoun is morphologically encoded into the verb form”) has been changed to: “[...] information about the gender and number of the predicate’s referent of a pronoun is morphologically encoded into the verb form” (p. 4, lines 71-73). In the second case, we provided an additional explanation of what a mismatch between a pronoun and a verb refers to: “that whenever a mismatch between a pronoun and a verb occurs (i.e. the pronoun refers to a less-preferred, non-topical or unexpected antecedent), the speaker needs to adjust the interpretation of the sentence and reidentify the referent of the anaphorical expression” (p. 4 line 80-82).

Example lines 66-67 needs a gloss

Response 2.15: A gloss has been added to all examples in Polish (including the examples of experimental sentences for Study 1 and examples from the Introduction).

Line 232: referent of a verb —> subject of a verb

Response 2.16: We respectfully disagree with this suggestion because in this particular case we do not assume that the referent of a verb would always be in the syntactic position of a subject. As we use the term “subject” to refer to the syntactic position within a sentence, we would not like to confuse the reader.

Fig 2: labels object-match and subject match are odd because they don’t relate to the experimental set up. Simply subject and object?

Response 2.17: The labels of Fig 2 were changed following your suggestion.

Lines 389-390: there are a lot of studies that have used eye-tracking to look at pronoun resolution, should refer to these.

Response 2.18: Thank you for pointing this out. A short review of the previous eye-tracking studies has been added to the text (p. 22, lines 522-528).

Line 382: higher fixation times: this is a prediction relating to a specific measure (fixation times), but you actually tested first-pass, go-past and total times which are more than simply fixation times.

Response 2.19: The prior formulation (“ [...] this cost should be reflected in higher fixation times in the critical region [...]”) was changed according to your suggestion (“[...] this cost should be reflected in longer reading times (measured by first-pass time, go-past and total-time) in the critical region [...]”, p. 21, line 512).

Line 404: total time is a cumulative, not a late measure (captures both early and late processes).

Response 2.20: The prior formulation (“This allowed us to interpret the processes engaged in sentence interpretation, which range from early (captured by the first-pass time) to late(captured by total time) measures [...]”) has been changed according to your suggestion (“This allowed us to interpret the processes engaged in sentence interpretation, which range from early (captured by the first-pass time) to cumulative (captured by total time) measures [...]”, p.23, lines 542-543).

Around line 415: I think I get what you mean in the predictions for the naturalness ratings, but you need to spell it out more clearly (what you mean by natural here).

Response 2.21: A more detailed explanation of the predictions has been added to the text. We modified the paragraph so that it would be clear that any assertions regarding the naturalness of a particular kind of sentence refers to our empirical data (i.e. natural = what was rated as most natural by the participants of Experiment 1):

“We expected the results of the naturalness rating to reflect the online anaphora resolution bias. Specifically, we expected that null pronoun sentences would obtain higher naturalness scores than overt pronoun sentences, and sentences following the preference for anaphora resolution (i.e., a null pronoun referring to the most-prominent antecedent and an overt pronoun referring to a less-prominent antecedent) would obtain higher scores than those violating it.” (p. 24, lines 577-582)

Condition examples (lines 464-475): need a gloss

Response 2.22: A gloss has been added to all examples in Polish (including the experimental sentence examples for Study 1 and an example from the Introduction).

Line 565, figure 4A caption is incorrect - not pronoun + verb (no pronoun because null)

Response 2.23: Indeed, the caption was incorrect. Thank you for spotting the mistake. We changed the previous formulation (pronoun + verb) to (Ø + verb). “Ø” is already used to refer to a null pronoun when examples of the sentences in the four experimental conditions of Study 1 and Study 2 are introduced (p. 17 lines 409-410, pp. 26-27 lines 634-643).

Line 635: syntactic position of an object —> syntactic position of a subject

Response 2.24: This has been corrected in the text.

##################

Reviewer #3: 

In my opinion, the present study presents several strong points: First, it is a close replication of a previous study (both in terms of tasks and materials); second, it adds a (Slavic) language to the repertoire of languages that have been studied in the pronoun resolution literature; third, the reported results confirm a well-known bias previously attested in other languages.

My concern regarding this article, as I explain just below, has to do with the fact that the literature review and parts of the discussion section are rather superficial (in particular when it comes to the discussion of the factors that play a role in pronoun resolution).

The present study replicates a previous study on the pronoun resolution preferences in Spanish (Chamorro et al., 2015). The authors of the present study set out to investigate the interpretation strategies for null and overt subject pronouns in another null-subject language, namely Polish. In order to do so, they make use of two experimental tasks, an offline sentence-interpretation task, where participants had to decide on the referent of an ambiguous pronoun and subsequently rate the naturalness of the sentence, and an online eye-tracking during reading task where participants read sentences, which they also had to rate for their naturalness afterwards. The main findings of the study are the following:

• Offline experiment: Polish speakers prefer to interpret the null pronoun as co-referential with the subject antecedent, and the overt pronoun as co-referential with the object antecedent. 

• Online experiment:

Finding #1: the interpretation of a null pronoun is not disrupted by a violation of its natural preference (for the subject antecedent); however, the interpretation of the overt pronoun is disrupted when it is forced to co-refer with the subject antecedent.

Finding #2: the sentences where the (null/overt) pronoun refers to the antecedent in the object position incur an additional processing cost as compared with the sentences where the pronoun refers to the subject antecedent. 

In my opinion, the present study presents several strong points: First, it is a close replication of a previous study (both in terms of tasks and materials); second, it adds a (Slavic) language to the repertoire of languages that have been studied in the pronoun resolution literature; third, the reported results confirm a well-known bias previously attested in other languages. 

My concern regarding this article, as I explain just below, has to do with the fact that the literature review and parts of the discussion section are rather superficial (in particular when it comes to the discussion of the factors that play a role in pronoun resolution). 

My recommendation is that the article is accepted after some revisions have been made.

These revisions have mainly to do with the discussion of the linguistic factors that play a role in pronoun resolution in the literature review and in the discussion/conclusion sections. The authors discuss in certain detail the issue of antecedent prominence as being crucial in pronoun resolution and, in relation to this, they discuss the syntactic function/position of the subject as being an important contributor of this prominence (this is reinforced by the discussion of Carminati’s Position of Antecedent Hypothesis and the related psycholinguistic findings on languages such as Italian and Spanish). However, the discussion stops there. No mention of other potential crucial factors (e.g. topicality, thematic role, order of mention, etc.) is to be found. The authors do evoke multiple times discourse-pragmatic factors (see the non-exhaustive list below of the sections where this comes up) but they never give examples of any of these factors nor do they discuss any previous studies that investigated them. 

Response 3.1: Thank you for your comment. Indeed, we fully acknowledge that our discussion of results was limited to the PAS framework (and generally to theories that propose syntactic preferences, especially subjecthood, as a dominant constraint on pronominal anaphora resolution). We agree that omission of other accounts and factors was a serious shortcoming of our argumentation, so we undertook several steps to fix this problem: 

1. We added a paragraph to the Introduction (p. 6 lines 111-126) describing different factors that have been shown to influence anaphora resolution (such as grammatical roles, order of mention, agentivity, topicality, focus, information structure and discourse coherence). We also included this information when discussing previous experimental evidence on anaphora resolution (p. 11, lines 248-255). We discussed how alternative accounts to PAS would translate into predictions for Study 1 (p. 16 lines 370-377) and we considered them when discussing the results of Study 1 and Study 2 (pp. 38-39 lines 900-915). 

2. Moreover, following the suggestions of Reviewer #2 regarding the review of the previous literature, we discuss Kehler and Rhode’s recent theory of pronominal anaphora resolution. This was proposed as a reconciliation of coherence-driven and centering-driven approaches (2013) (pp. 9-10 lines 211-234) and it differentiates between “hard” morpho-syntactic constraints of pronoun reference (such as gender, number, person etc.) and “soft” constraints (such as grammatical or thematic roles, syntactic function, coherence etc.). The discussion of our result also reflects this proposal (p.41, lines 976-980; p. 43 lines 1012-1016). 

3. To better explain what discourse-related factors refer to, we made several adjustments throughout the text. We now refer to two distinctions proposed in previous literature which describe factors that influence pronominal anaphora resolution: (1) “hard” morpho-syntactic constraints (e.g. number, gender, person etc.) vs. “soft” constraints or heuristics (such as grammatical or thematic roles, syntactic function, coherence etc.), which were introduced and discussed by Kehler and Rhode (2013) (pp. 9–10 lines 211–234); (2) syntactico-semantic factors (subjecthood and grammatical roles, thematic roles, agentivity) vs. discourse-related factors (topicalization, focus, order of mention), which were introduced by Kaiser and Trueswell (2008) (p. 6 lines 111-126). This allowed us to discuss our results within the existing frameworks, extend our predictions and findings beyond the syntax-based constraint of subjecthood, as well as better define the “discourse-related” factors. All mentions of syntax-based cues have been changed accordingly throughout the text

The discussion of these additional factors is paramount in order to be able to better account for the present findings. This seems to be the case regarding the finding that the sentences where a (null/overt) pronoun refers to the antecedent in the object position incur an additional processing cost compared with the sentences where the pronoun refers to the subject antecedent, which the authors account in the following terms: “This suggests that Polish speakers interpret sentences under the working assumption that they refer to the most prominent subject antecedent” (p. 27, line 640-645). What we see in this case (i.e. in the object conditions) is a shift in the sentence topic from the main clause to the subordinate clause. This dispreference for a topic-shift intra-sententially has also been attested in other languages, such as French, German, and Spanish (see Colonna et al., 2012, 2015; de la Fuente, 2016; Patterson & Felser, 2020). The notion of (sentence) topic and these previous findings need to be discussed in the paper.

Response 3.2: Thank you very much for pointing out these very valuable references. We added a fragment to the General Discussion in which we discuss our results in the light of anti-focus effects (pp. 44-45 lines 1056-1063). Additionally, a mention of the early and late stages of the pronoun resolution process that was discussed by Patterson & Felser (2020) has been included in our Discussion (p. 44, lines 1041-1044). Following other comments from all the Reviewers, to the Introduction (p. 6, l. 119–121) we added a more in-depth discussion of factors which can influence the anaphora resolution process, including topicality and focus. Mentions of topicality as an important cue of antecedents’ prominence appear throughout the entire text.

To account for the information presented in the discussion of anti-focus results in the context of our study, the sentence quoted in the current comment was rephrased to the following (no additional explanation has been added as a discussion of the anti-focus effect appears later in the same paragraph):

“This suggests that Polish speakers interpret sentences under the working assumption that the topic of a sentence is the most-prominent, first-mentioned, subject antecedent” (p. 37 lines 858-860).

My second (more general) remark concerns the choice of manipulation regarding the disambiguating part of the experimental sentences in the online experiment, namely the agreement between the subject of the main clause and the verb of the subordinate clause (a manipulation also used by Chamorro et al. 2015 in their experiment in Spanish). The pronoun resolution strategies commonly observed in the literature (null pronoun refers to a subject antecedent, overt pronoun refers to an object antecedent) remain preferences which can be easily overridden by other factors (this could indeed explain why there is certain variability in the literature regarding the resolution preference for Spanish null and overt pronouns, for example). Morpho-syntactic cues such as gender and number agreement between pronoun and antecedent and subject-verb agreement are examples of such strong cues which can override the role of other factors in pronoun resolution (like the position of the antecedent). That is why the differences between conditions 1 and 2, on the one hand, and conditions 3 and 4, on the other hand, are not always robust.

Response 3.3: We fully acknowledge the Reviewer's concern and agree with the remark regarding some problematic aspects of the used manipulation. Since our aim was to replicate as closely as possible the initial study by Charomorro, we used methodology that was as similar as possible. Since data collection is already completed, there is nothing that can be changed in the design at this point. In future research, we will certainly take this remark into account when exploring new avenues of research on pronoun resolution in Polish. Still, despite the fact that our study is certainly not free of limitations, we believe it contributes to the body of knowledge on pronoun resolution mechanisms in different languages.

We hope that the concerns raised by the Reviewer can – as much as is possible at this stage – be addressed by including an additional explanation of the role of “hard” morpho-syntactic cues (i.e. subject-verb agreement) and the “soft” cues that can override the syntactically-based preferences in pronoun resolution. These additional explanations are mainly related to a discussion of the additional theoretical account by Kehler and Rhode (2013), which was suggested by Reviewer #2 (R2.1) (pp. 9–10 lines 211-234).

In relation to this, I was not very surprised to read that the interpretation of a null pronoun is not disrupted by a violation of its natural preference (for the subject antecedent); however, the interpretation of the overt pronoun is disrupted when it is forced to co-refer with the subject antecedent (contrary to what was previously found in the literature). For me, there is a difference in terms of acceptability between the two “less-natural” conditions 1 and 4: Condition 1 (overt pronoun/subject match) seems a lot less natural than condition 4 (null pronoun/object match). This has to do, on the one hand, with what I described in the paragraph above regarding subject-verb agreement and, on the other hand, with the fact that the overt pronoun is a lot more (pragmatically) marked and not always pragmatically felicitous in short decontextualized sentences. 

Mention of syntax vs. pragmatic/discourse factors: 

(p. 2) “Overall, in the Polish language, interpretation of a null pronoun seems to be more sensitive to pragmatic cues of reference than syntactic cues of reference, while resolution of overt pronouns relies strongly on syntax-based cues.” 

Response 3.4: Following the reformulation of the text such that it would not be based on syntax-based vs. pragmatic factors, this sentence has been deleted from the abstract.

(p. 3) “their reference relies not only on the rigid rules of the syntax of a given language but is also sensitive to pragmatic and contextual factors”

Response 3.5: This fragment has been replaced with the following more general statement: “As a consequence, pronouns are inherently ambiguous: their reference depends on a range of different constraints, including morpho-syntactic, semantic and discourse-related cues”

(p. 3) “the mechanisms of anaphora resolution can be sensitive to syntactic and also pragmatic or contextual factors” 

Response 3.6: This fragment has been replaced by the following: “The mechanism of anaphora resolution can be influenced by a range of factors which affect different stages of pronoun interpretation: from syntactico-semantic processing to constructing mental representations of discourse.”

(p. 4) (discourse-oriented language) “other sources of information such as discourse and context become much more important for the correct resolution of an anaphoric expression”

Response 3.7: In this case, the reference to discourse as a source of cues that drive the anaphora-resolution process is a reference to the paper by Kwon and Sturt (2013), to which we refer when introducing sentence-oriented and discourse-oriented languages: “in so-called discourse-oriented languages without such a verbal agreement system, like Japanese, Korean, and Chinese, discourse may play a stronger role.” (Kwon & Sturt 2013, p. 379). As such, we would like to retain the original terminology; however, following your comment, we have attempted to propose a rewording of the fragment in question. As we previously mentioned, we have also included a more extensive explanation of what discourse-related cues of reference are in a further part of the text (p. 6, lines 119-126). 

The fragment in question and the sentence directly preceding it have been replaced with the following: “In contrast to sentence-oriented languages, discourse-oriented languages which do not have a verbal agreement system, such as Japanese, Chinese or Korean, rely much less (or not at all) on morpho-syntactic constraints (4). Therefore, other cues of reference that are derived from discourse become much more important for the correct resolution of an anaphoric expression (3,4).” (pp. 4–5, l. 83–87).

(p. 6) “differences between languages that are driven by language-specific, syntax-based, pragmatic constraints” 

Response 3.8: This fragment has been replaced with the following: “differences between languages that are driven by language-specific sensitivity to different constraints on pronoun resolution”

(p. 10, line 216) “stem from divergent sensitivity of null and overt pronouns to syntactic and discourse-related or context-related determinants of the prominence of an antecedent” 

R3.9 Response: This fragment has been replaced by the following: “[...] stem from the divergent sensitivity of null and overt pronouns to cues of prominence of antecedents which can be driven by syntactico-semantic or discourse-related constraints”. 

Even though our reformulation does not dispose of the problematic distinction but replaces it with another (i.e. syntactico-semantic vs. discourse-related constraints), we believe that after providing a more elaborate explanation of what factors have been shown to affect the anaphora-resolution process (e.g. subjecthood, topicality, agentivity etc.) and explaining the distinction between syntactico-semantic and discourse-related factors (p. 6 lines 111-126), it should be clear to the reader what we refer to here.

(p. 11, line 251) “can be prone to the influence of other non-syntactic factors. These factors…”

Response 3.10: This fragment has been replaced with: “[...] can be prone to the influence of other cues of prominence of the antecedent. These factors [...]”

(p. 16, line 387-388) “we should be able to assess the sensitivity of null and overt pronouns to the syntax-based determinants of the prominence of an antecedent”

Response 3.11: As a consequence of modifications introduced to address another Reviewer’s comment, this fragment was deleted.

(p. 17, line 398-399) “can provide information on the sensitivity of pronouns to different types of prominence cues”

Response 3.12: This fragment was deleted following the suggestions of the Reviewers who pointed out that our design did not make it possible to test the sensitivity of pronouns to different cues of prominence. As we fully agree with this point, we have changed the narration to avoid any misleading claims which might suggest that we tested the sensitivity of pronouns to different types of prominence cues.

(p. 29, line 708-710) “relies on both syntactic and contextual factors”[…] “can be modulated by pragmatic and contextual cues” 

R3.13 Response: This fragment has been replaced (following a broader change to our narration) with the following: “So far, we have argued that subjecthood, order of mention and automatic assignment of topicality to the subject-antecedent are the dominant constraints on anaphora resolution in Polish, which in some cases can be overridden to better accommodate a current representation of discourse.”

(p. 8-9) “Whenever a violation of a preferred pronoun-antecedent match incurs additional processing cost, this indicates that the speaker is using a syntactic cue (i.e. the antecedent’s syntactic position within a sentence) to identify the referent of an anaphorical pronoun. On the other hand, if violating the speaker’s preference does not lead to additional processing cost, a speaker likely uses non-syntactic cues to define the prominence of an antecedent (i.e. context-related or discourse-related information) to interpret the anaphorical pronoun.” 

R3.14 Response: This fragment was deleted following comments from all the Reviewers, who pointed out that, based on our design and the existing literature, we do not have a basis for this claim. We agree with this, so we decided to remove this paragraph from the predictions and instead elaborate more on possible factors driving the anaphora resolution mechanism in Polish in the Discussion (see pp. 38–39 lines 900-915).

• What context-related or discourse-related information? It seems that “discourse” and “contextual” are used as synonymous terms in certain sections of the paper, and as two different things in other (e.g. on page 28, the use of the term “context-dependent information” seems to refer to the information on subject-verb agreement). 

Response 3.15: This is true. We agree that we were not very clear about what discourse-related information is and how it relates to context (and we have in fact used these two terms as synonyms in several instances). We have added an explanation of what discourse-related factors mean (p. 6 lines 119–126), and we avoid mentions of “context-dependent information” throughout the text to avoid confusion. As for the example that you mention (p. 28), we deleted the sentence mentioning “context-dependent information” as, after introducing other changes suggested by all the Reviewers, this fragment was unnecessary and unclear.

• Are there previous findings that show this clear-cut tendency? Is this related to the findings for Italian and Spanish by Filiaci and colleagues? Because there is crosslinguistic variation there and even variation within the same language (Spanish) if we take into account the results by Chamorro and colleagues.

Response 3.16: As mentioned in the response to the general comment, we acknowledge we have no grounds for making these kinds of claims, therefore we have deleted the problematic fragment and we discuss the possible factors that might influence pronoun resolution without making such strong categorical distinctions.

• (p. 10, line 213) “the available empirical evidence demonstrates that the preference for anaphora resolution seems to be more universal for null pronouns” > taken into account the two languages under discussion in the literature review, this is not so clear cut, right? It seems to be the case for Italian, for not so much for Spanish. 

Response 3.17: Thank you for bringing this part to our attention. In the context of the paragraph it was located in, it might have been easy to misinterpret as this particular claim actually refers to only one of the discussed experiments, namely that of Filiaci et al. (2014). What we meant to say was that, despite the differences between the two experiments testing pronouns’ sensitivity to violation of their preferences (that might be due, e.g., to the experimental stimuli used), the differences in pronoun resolution strategies between the two languages only emerge for overt pronouns, not null pronouns.

We have rephrased the preceding paragraph as well as the conclusion (from which the statement you brought up in your comment was taken) to clarify what we mean (p.12 lines 287-302):

“However, no differences in the sensitivity of null and overt pronouns to the violation of their preferences was observed for Spanish. Differences in whether the processing penalty is observed for sentences with null or overt pronouns referring to their less-preferred antecedents can be accounted for by differences in the design between the studies by Chamorro and colleagues and Filiaci and colleagues (in particular, by the clause order: the stimuli in Chamorro and colleagues’ study (21) followed the Main-Subordinate order, while in Filiaci and colleagues’ study (5) the order was Subordinate-Main. For further discussion, see Discussion). Interestingly, despite the discrepancies between the results of these two experiments, the results of the experiment by Filiaci and colleagues (5) demonstrate that a preference for null pronoun resolution is similar for Italian and Spanish, while differences emerge for overt pronouns.

To conclude, the cross-linguistic differences in anaphora resolution might stem from the divergent sensitivity of null and overt pronouns to cues of prominence of antecedents, which can be driven by syntactico-semantic or discourse-related constraints (1,24)”

Other comments 

(p. 4) sentence-oriented languages (e.g. Italian, Polish, Spanish) rely on morpho-syntactic cues (subject-verb agreement) to establish the referent of an anaphorical expression. > Nothing about sentence vs. discourse oriented languages in the discussion (does Polish behave like a hybrid (sentence-oriented for null pronouns) but discourse-oriented for overt pronouns)? 

Response 3.18: The distinction between sentence-oriented and discourse-related subjects is mainly based on the morphological characteristics of a language, namely whether it has a subject-verb agreement or not (i.e., whether a pronoun can be resolved based on the morphological characteristics of a verb). As such, Polish is an inherently sentence-oriented language. We introduced the distinction between sentence-oriented and discourse-oriented languages at the beginning of the discussion to draw attention to the variety of different cues that different languages use to establish the reference of a pronoun. However, we acknowledge that introducing this distinction needs addressing in further parts of the text, so we have added the following information:

Introduction of the coherence-driven approach to pronoun resolution:

“Their theory differentiates between “hard” constraints of reference, which in the case of sentence-oriented languages relate to the morpho-syntactic features that determine the subject-verb agreement (such as gender, number, person etc.) […]” (p.10, lines 227-230)

Discussion:

“The results of both reported studies suggest that subjecthood and information structure (i.e., order of mention but also topicality and thematic roles) are dominant constraints for pronoun resolution in Polish” (p. 39 lines 911-913).

(p. 3) The assertion that the pronoun she and the noun Mary in When Mary crossed the street, she looked back at the monument “denominate one and the same person” is misleading. Strictly speaking, she can also refer to somebody else other than Mary here. 

Response 3.19: Yes, we agree that “she” could also refer to somebody else, but for the sake of the example we present we’d like to keep it simple and not include additional explanations of other referential possibilities in the text. However, we have rephrased this fragment so that it does not make such a strong claim:

“the reader needs to know that “she” in the second clause refers to “Mary” in the first clause and both words are supposed to denote one and the same person”

(p. 4, 14, 20) Given the rich morphology of the language, the examples in Polish should also include a gloss in English.

Response 3.20: A gloss has been added to all examples in Polish.

(p. 5, line 90) “on the other hand, less-reduced expression, like those containing an overt pronoun” > “on the other hand, less-reduced expression, like an overt pronoun” 

Response 3.21: Thank you. The suggested change has been introduced in the text.

(p. 14, line 338-340) Did the filler items also feature null and overt pronouns? If so, I’m not sure I’m very comfortable with the fact that the filler items also included the critical (null vs. overt pronoun) manipulation.

Response 3.22: Filler items did not include the critical manipulation, even though they included null and overt pronouns. The filler sentences are similar to the experimental sentences only in terms of length and the fact that they are composed of Main and Subordinate clauses (but not necessarily in the Main-Subordinate order, as in the case of the experimental sentences). As Polish is a pro-drop language, null pronouns appeared in most of the sentences (as pronouns are usually skipped before verbs). In some sentences, we also used overt pronouns, but they were either non-anaphoric (e.g., “On zgodził się, że powieści Sienkiewicza nie są ciekawą lekturą dla nastolatków.” – He (overt pronoun) agreed that the novels by Sienkiewicz are not an interesting read for teenagers) or they were unambiguous. Crucially, there were no sentences in which null or overt pronouns were forced to refer to a less-preferred antecedent.

If this was the case, I was also wondering whether the filler items also contained the same proportions of null and overt pronouns. This is an important point, as it was been previously shown that the varying proportions of the type of pronoun present in the experiment has a direct impact on the participants’ resolution preference (cf. Fernandes et al. 2018). 

Response 3.23: Thank you for pointing out this important work that shows adaptation effects. We did not take it into account when designing our experiment and the filler items; as the data collection is already concluded, there is not much we can do about the stimuli choice; however, we believe that adaptation effects might not be very problematic in the case of our study. As we pointed out in the previous comment, due to the characteristics of Polish, overt pronouns are much less common than null pronouns. As such, overt pronouns appear in the filler sentences less often than null pronouns; however, they were never used in an ambiguous way, they were not anaphoric pronouns, and they were never forced to refer to less-preferred antecedents. Moreover, the structure of the filler sentences was different from that of the experimental sentences (see the examples below). We believe that this might be an important point as the effect of adaptation in pronoun resolution that was shown in the study by Fernandes et al. (2018) was reported based on an experiment which used a set of sentences of identical structure (subject – verb – object – conjunction “when” – pronoun (null or overt) – verbs – complement). As such, it might be possible that it is driven at least to some extent by syntactic priming, similarly to grammatical parallelism bias (Smyth, 1994). As such, using filler sentences with different grammatical structures might actually reduce the adaptation effect.

A couple of examples of filler items that we used in our study (overt pronouns are marked with *):

1. Ona lubi malować wieczorami, podczas gdy on gra na gitarze i śpiewa.

She* likes to paint in the evenings while he* plays the guitar and sings.

2. Oni muszą skończyć remont, zanim wyprowadzą się z tego mieszkania do nowego domu.

They* need to finish the renovation before they move from this flat to a new house.

3. Podczas gdy oni podróżowali po Afryce, ich kuzynka urodziła śliczne bliźnięta.

While they* were travelling in Africa, their* cousin had lovely twins.

4. Pingwiny często gromadzą się blisko siebie, żeby było im cieplej.

Penguins often gather close together so that they are warmer.

I’m also little concerned about the use of the different conjunctions for the filler sentences. We know from previous literature (the works by Kehler and Rhode, de la Fuente et al. 2016), that the kinds of coherence relations that these types of conjunctions establish between main and subordinate clause, have an impact on pronoun resolution. I wonder to what extent the filler items also influenced the resolution preference of the experimental items here.

Response 3.24: This is a very interesting question, especially in light of the adaptation account discussed in the previous comment. However, due to the variability in the conjunctions and the structure of the filler sentences, we are not able to statistically estimate this effect based on our data. In the case of our experimental stimuli, the decision to use a range of different conjunctions and grammatical structures in the filler sentences was driven by the fact that we based our design on the study by Chamorro et al (2016); as such, we attempted to closely follow their solutions. However, as pointed out in the previous comment, our filler sentences did not exactly follow the same grammatical structure as the experimental stimuli; as such, the biases in pronoun interpretation induced by the use of different conjunctions should not have primed the pronoun resolution preferences in the experimental sentences (or, at least, there should not have been a simple syntactic priming between the filler and experimental sentences).

(p. 15, line 344) “was modulated by the use of a pronoun” > “was modulated by the nature of the pronoun” 

Response 3.25: Thank you. The suggested change has been introduced in the text.

(p. 15, line 364-365) “which indicates that native Polish speakers find null pronoun sentences more natural than overt pronoun sentences” > This sounds a little too general. A better formulation would be “which indicates that native Polish speakers find the null pronoun condition more natural than the overt pronoun conditions” or “which indicates that native Polish speakers find the sentences containing a null pronoun more natural than those containing an overt pronoun” 

Response 3.26: Thank you. The suggested change (the second reformulation that was proposed in your comment) has been introduced in the text.

(p. 17, line 406-412) Why would this be your prediction? As you said previously, the preference of the null pronoun for a subject antecedent seems to be more robust according to previous findings. The findings of Chamorro et al. are rather exceptional in that respect. Shouldn’t you then expect the opposite (i.e. that null pronouns are more sensitive to violations of the syntax-based bias)?

Response 3.27: Thank you for raising this issue. A similar comment was also made by Reviewer #1 (R1.6). We agree that the reason why we would make this prediction is not straightforward. We provided a more in-depth explanation in the text that refers to the results of the naturalness judgement task from Study 1 (pp. 23-24, lines 546-571). In the General Discussion (pp. 41-42, lines 1000-1016), we also address this issue with respect to experiments that show a contradictory pattern of results. 

(p. 20, line 476-480) It is not clear to me whether there were 8 experimental items (in the 4 experimental conditions = 32 experimental sentences in total = 4 lists each with 8 experimental items and 64 filler items) or 32 experimental items (in the four experimental conditions = 128 experimental sentences in total = 4 lists each with 32 experimental items and 64 filler items). 

Response 3.28: Thank you for drawing our attention to this ambiguity. We have updated the description of the experimental lists in the text in the following way to make it clearer:

“[...] four different experimental lists of 32 sentences were created. Each list contained eight sentences corresponding to each of the four conditions (i.e., 8 sentences in Condition 1, 2, 3 and 4). Additionally, 64 filler sentences were created and added to the experimental lists. They contained different grammatical structures, inanimate referents, proper names and plural pronouns. Each of the lists contained the same 64 filler sentences. In total, each experimental list contained 96 sentences (32 experimental + 64 filler sentences).” (p. 27, lines 643-650)

(p. 21, line 509) Strictly speaking, the go-past measure is more of a “hybrid” measure that reflects both early and late processes. 

Response 3.29: Thank you for pointing this out. We have changed the text according to your suggestions by replacing the previous formulation: “The first two measures are presumed to be sensitive to processes that occur relatively early in sentence comprehension” to the following: “First-pass time is presumed to be sensitive to processes that occur relatively early in sentence comprehension” (p. 28 lines 737–738) and adding a sentence explaining the go-past measure: “Go-past time is a hybrid measure that is sensitive to both early and late processes in sentence comprehension” (p. 28, lines 743–745).

(p. 26, summary and general discussion sections) The authors use multiple times formulations such as “processing sentences which do not refer to the most prominent antecedent” or “sentences referring to an antecedent in the object position always incur…” This sounds a little weird. Strictly speaking, it is the pronoun that refers to one or another antecedent not the sentence. A better formulation can be “sentences in which the (null/overt) pronoun refers to…”. 

Response 3.30: Thank you for spotting this problem and suggesting a solution. We have corrected all occurrences of the problematic formulation in the Summary and General Discussion.

---

## [Decision Letter · Decision Letter 1]

22 Sep 2021

PONE-D-21-12740R1Pronominal anaphora resolution in Polish: investigating online sentence interpretation using eye-tracking.PLOS ONE

Dear Dr. Wolna,

Thank you for submitting your manuscript to PLOS ONE. After careful consideration, we feel that it has merit but does not fully meet PLOS ONE’s publication criteria as it currently stands. Therefore, we invite you to submit a revised version of the manuscript that addresses the points raised during the review process. Your revised manuscript was sent to two of the original reviewers for re-evaluation. Although both are happy with the way you have addressed their initial concerns, they both list a number of minor points which should be addressed in the final version of your manuscript.

We look forward to receiving your revised manuscript.

Kind regards,

Claudia Felser, Ph.D

Academic Editor

PLOS ONE

Journal Requirements:

Additional Editor Comments (if provided):

Please also reword any unclear, misleading, or stylistically convoluted passages in your manuscript. Examples of these can already be found in your abstract:

"cues of reference" - odd wording

"an ambiguous sentence interpretation task" - this phrase is itself ambiguous

"First, in an ambiguous sentence interpretation task, we explored the natural biases that occur during the interpretation of null or overt pronouns." - What is meant by "natural" here (are there also unnatural biases)? Also, if this was an offline task, how can it possibly explore biases *during* interpretation?

"antecedents which are in the syntactic position of a subject or an object" - Why not simply say "subject or object antecedents"?

Please carefully check the remainder of your manuscript for further such instances and amend these. Reviewer #1 makes several helpful suggestions in this regard.

Reviewers' comments:

Reviewer's Responses to Questions

**Comments to the Author**

1. If the authors have adequately addressed your comments raised in a previous round of review and you feel that this manuscript is now acceptable for publication, you may indicate that here to bypass the “Comments to the Author” section, enter your conflict of interest statement in the “Confidential to Editor” section, and submit your "Accept" recommendation.

Reviewer #1: (No Response)

Reviewer #2: (No Response)

2. Is the manuscript technically sound, and do the data support the conclusions?

Reviewer #1: Yes

Reviewer #2: Yes

3. Has the statistical analysis been performed appropriately and rigorously? 

Reviewer #1: Yes

Reviewer #2: Yes

4. Have the authors made all data underlying the findings in their manuscript fully available?

Reviewer #1: Yes

Reviewer #2: Yes

5. Is the manuscript presented in an intelligible fashion and written in standard English?

Reviewer #1: Yes

Reviewer #2: Yes

6. Review Comments to the Author

Reviewer #1: I was reviewer 1 on a previous draft of this manuscript. My previous major concerns have been addressed satisfactorily, and I do not have any large objections, though I do have some comments on this revised version and some additional minor suggestions.

Comments:

- p. 9, l. 191-197: “For example, in the case of overt pronoun interpretation in Italian and Spanish, the increased processing cost resulting from forcing a shift of reference from the syntactic subject of a sentence (which is also the first-mentioned referent) to the syntactic object is observed for Italian but not for Spanish. Within the Form-Specific Multiple-Constraint Approach, this cross-linguistic difference can be explained by referring to the different sensitivity of pronouns to syntactico-semantic cues in both these languages: in Italian, pronoun resolution seems to rely more heavily on this type of cue than in Spanish.” But why are Italian overt pronouns more sensitive to syntactico-semantic cues than Spanish overt pronouns? “differences in pronouns’ reference resolution do not necessarily need to be associated with typological differences between languages” (l.186-187): this seems to be the case instead if the cross-linguistic difference in overt pronouns interpretation depends of the different sensitivity of overt pronouns in Italian and in Spanish. “differences between sentence-oriented and discourse-oriented languages can be accounted for by referring to the much higher sensitivity of pronouns to the informational structure of discourse in discourse-oriented languages than in sentence-oriented languages.”(l. 198-200): Here again, typological differences between languages are seen to be responsible for cross-linguistics differences in pronoun interpretation.

- Thematic roles are defined as the semantic roles (agent, patient,…) (see p. 6, l. 109-111). However, further on in the text, I do not understand the formulations: “the role of the thematic topic” (p. 40, l. 945) “and the thematic role of a topic” (p.41, l. 967). I would change to “the role of topic”. Furthermore, the authors propose to distinguish the syntacto-semantic cues from discourse-related cues but in the conclusion, they seem to include topicality in the syntacto-semantic cues (p. 41, l. 977) when I understood that they were talking about the discourse topic throughout the paper. This should be clarified.

- In the discussion (p. 37-38), the authors discuss the role of the context of a sentence but again, the context was not manipulated in the current study. So, contrary to what the authors claim, I do not think that the online results allow them “to determine whether these preferences can be modulated by the context of a given sentence” (p. 37, l. 858). Just as I do not think that they can conclude that null pronouns “easily shift their reference towards a more contextually appropriate referent” (p. 38, l. 888) since null pronouns, whether disambiguated in favour of the object or the subject, were presented in the same contexts.

- In further discussion (p. 41-42), the argument about the anti-focus effect needs clarification. To the best of my knowledge, contrary to what the authors claim (p. 40, l. 952-953), an increased processing cost associated with an anti-focus effect was not observed. The anti-focus effect is the fact that the pronoun refers preferentially to the topic of the current discourse unit (which is not the focused by cleft antecedent of the discourse unit in which it appears since the clefting function is to signal a topic shift for the subsequent discourse unit). The authors should clarify what they want to argue here.

Smaller suggestions:

- p. 2, l. 19: in the abstract, change “we explored the natural biases that occur during the interpretation of null or overt pronouns” to “we explored the interpretation preferences of null and overt pronouns”

- p. 9, l. 196, change “pronoun resolution seem” to “pronoun resolution seems”

- p. 10, l. 235, change “the natural pronoun-antecedent match” to “the preferred pronoun-antecedent match”

- p. 11, l. 247, change “the natural anaphora resolution preference” to “the preferred anaphora interpretation”

- p. 11, l. 255, why is “followed” in italics?

- p. 11, l. 256, change “a violation of the natural preference” to “a violation of the offline preference”

- p. 13, l. 294, change “neither the preference” to “neither the preferences”

- p. 14, 332, in the title, change “What is the natural pattern of anaphora resolution in Native Polish speakers?” to “What is the offline pattern of anaphora resolution in Native Polish speakers?”

- p. 16, l. 366-367, change “In order to test whether a pronoun can influence the natural tendency to interpret the antecedent of a sentence” to “In order to test whether the pronoun form (null vs overt) can influence its interpretation preference”

- p. 20, l. 477, change “violation of the natural preference for anaphora resolution” to “violation of the preferred anaphora interpretation”

- p. 27, l. 635, change “on native Polish speakers’ natural preference” to on native Polish speakers’ offline preference”

- p. 33, l. 758, change “the natural pattern of pronominal anaphora resolution in Polish” to “the pattern of pronominal anaphora resolution in Polish”

- p. 33, l. 759, change “we established a preference” to “we established the offline preference"

- p. 33, l. 768, change “the interpretation of a null pronoun is not disrupted by a violation of its preference” to “the interpretation of a null pronoun is not disrupted by a violation of its subject-preference”

- p. 34, l. 777-779, change “Polish speakers interpret sentences under the working assumption that the topic of a sentence is the most-prominent, first-mentioned, subject antecedent” to “Polish speakers interpret sentences under the working assumption that the most-prominent referent is the first-mentioned, subject and topic antecedent”

- I did not find Appendix A

Reviewer #2: I think that this paper has improved substantially since the previous version, I appreciate the efforts to which the authors have gone to address all comments in the last round. My remaining comments are quite minor. The area that now requires a bit of refinement is the discussion, as you will see from my final few comments.

Page 4 line 69: this example needs a number and a translation

Page 4 line 74: I wouldn’t describe this as a mismatch between a pronoun and a verb. It is a bit confusing to describe it this way, although I know what you are getting at. Rephrase. I think this issue is made more confusing by the fact that, up to this point in the manuscript, you haven’t indicated exactly what information the pronoun carries in Polish (also this is not clear from the example gloss). This should be clarified.

Page 10, lines 217: note that the SMASH algorithm is what Kehler and colleagues are arguing **against**

See also page 38 line 880.

Page 15, line 343: I think there is some kind of paste/text error here

Page 15, line 352-354. Without knowing the contents of the Alonso-Ovalle or Chamorro studies, it is very unclear why sentences with null pronouns should be overall more natural than those containing overt pronouns. Explain.

Page 16, examples: number the examples, and give a translation as well (it is normal to present both a gloss and a translation)

Page 19, statistical reporting: check if the term “b coefficient” is the correct one for this type of statistical analysis. I think what you normally get from a Bayesian ordinal model is an effect size and a credible interval. Check that you are reporting this correctly.

Page 19, line 446-447: “subject-matching sentences were rated as more natural than object-matching sentences for sentences containing overt pronouns. But in Table 1, the mean rating for subjects is much lower than for objects.

Page 20, study 1 summary: add some nuance to this paragraph. It sounds here as though you are describing categorical patterns (null pronoun only interpreted as subject and overt only as object), which is not the case.

Page 25: provide a translation for the example materials

Page 32, line 724: “sentences referring to object antecedent were rated on average as more natural than sentences referring to the subject antecedent”. You can ignore this main effect of antecedent here, as it interacts with pronoun - so the main effect of antecedent is driven entirely by the difference in the object pronouns. (In general, if main effects are qualified by a significant interaction they should not be interpreted on their own)

Discussion section: I suggest condensing this, it is quite long and in some places repeats the same point in several ways.

Pages 39-40: spillover effect and lexical access. It is difficult to see how this relates directly to the current study. The studies you cite are talking about why effects show up at the spillover region at all (as opposed to a region containing a particular stimulus). But the question you need to answer is why your spillover effect goes in a particular direction. I think this is an entirely different question, so I suggest cutting out this short part of the discussion and references 49-51.

Page 40: you should not make it sound as though Patterson & Felser were the first to propose that pronoun processing occurs in stages over time, this idea has been around in the literature for a very long time

Pages 40-41: I think you have misunderstood the anti-focus effect: it does not represent a cost associated with a “less prominent, non-focused antecedent”

7. PLOS authors have the option to publish the peer review history of their article (what does this mean?). If published, this will include your full peer review and any attached files.

Reviewer #1: No

Reviewer #2: No

---

## [Author Response · Author response to Decision Letter 1]

26 Sep 2021

We would like to thank the Editor and the Reviewers for their detailed comments and suggestions. We hope that the revisions in the manuscript and our accompanying responses successfully address the concerns of the Editor and the Reviewers and that after revision you will find our work suitable for publication in PLOS One.

Whenever in our responses we refer to specific pages and lines corresponding to the changes that we have introduced to the manuscript, the page and lines numbers refer to the marked-up version of the manuscript.

Yours sincerely,

Agata Wolna, Joanna Durlik, Zofia Wodniecka

Response to the Editor’s comments:

Please reword any unclear, misleading, or stylistically convoluted passages in your manuscript. Examples of these can already be found in your abstract:

"cues of reference" - odd wording

"an ambiguous sentence interpretation task" - this phrase is itself ambiguous

"First, in an ambiguous sentence interpretation task, we explored the natural biases that occur during the interpretation of null or overt pronouns." - What is meant by "natural" here (are there also unnatural biases)? Also, if this was an offline task, how can it possibly explore biases *during* interpretation?

"antecedents which are in the syntactic position of a subject or an object" - Why not simply say "subject or object antecedents"?

Please carefully check the remainder of your manuscript for further such instances and amend these. Reviewer #1 makes several helpful suggestions in this regard.

Response:

We have reviewed the manuscript and fixed all the unclear, misleading and (statistically) convoluted passages that we could identify. In the case of the examples that you mentioned in your comment:

• We reworded the formulation “cues of reference” wherever it appeared in this form to “factors determining the reference/prominence”; however the term “cues” (e.g. “discourse prominence cues”) is commonly used in the pronominal anaphora resolution literature so we decided to keep the word “cues” did not correct it in other instances

• We fixed the ambiguous phrase (“an ambiguous sentence interpretation task”) in the abstract. In the current version of the text it is as follows:

“we explored the interpretation preferences of null and overt pronouns in an ambiguous sentence interpretation task”

• We have replaced the more precise terms “antecedents in the syntactic position of subject/object” with the shorter version that you propose: “subject/object antecedent”. In a couple of instances we did not change the wording which stresses the syntactic position of an antecedent, but only if it was justified by the narrative(e.g. on p. 40 l. 953-954, where we believe that it is crucial to stress that the subjecthood/objecthood refers to a syntactic feature of an antecedent, because we build an argument that the unexpected processing cost observed in the post-critical AOI is actually related to the syntactic processing)

• We have incorporated in the text all other stylistic suggestions of Reviewer #1

 

Response to the Reviewers’ comments

Reviewer #1

I was reviewer 1 on a previous draft of this manuscript. My previous major concerns have been addressed satisfactorily, and I do not have any large objections, though I do have some comments on this revised version and some additional minor suggestions.

Comments:

- p. 9, l. 191-197: “For example, in the case of overt pronoun interpretation in Italian and Spanish, the increased processing cost resulting from forcing a shift of reference from the syntactic subject of a sentence (which is also the first-mentioned referent) to the syntactic object is observed for Italian but not for Spanish. Within the Form-Specific Multiple-Constraint Approach, this cross-linguistic difference can be explained by referring to the different sensitivity of pronouns to syntactico-semantic cues in both these languages: in Italian, pronoun resolution seems to rely more heavily on this type of cue than in Spanish.” But why are Italian overt pronouns more sensitive to syntactico-semantic cues than Spanish overt pronouns? “differences in pronouns’ reference resolution do not necessarily need to be associated with typological differences between languages” (l.186-187): this seems to be the case instead if the cross-linguistic difference in overt pronouns interpretation depends of the different sensitivity of overt pronouns in Italian and in Spanish. “differences between sentence-oriented and discourse-oriented languages can be accounted for by referring to the much higher sensitivity of pronouns to the informational structure of discourse in discourse-oriented languages than in sentence-oriented languages.”(l. 198-200): Here again, typological differences between languages are seen to be responsible for cross-linguistics differences in pronoun interpretation.

Response 1.1: Thank you very much for spotting this problem. In the fragment that the Reviewer found unclear, what we wanted to argue was that sensitivity to the syntactico-semantic cues may account for differences in pronoun resolution preference even between languages that are not typologically distant (i.e.: Spanish vs. Italian, in contrast to e.g. Spanish vs. Chinese). However, we agree with the Reviewer that different sensitivity to some types of cues can actually be considered to be a typological difference between the languages and as such our argument was somewhat circular.

As the considerations of typological distance between Spanish and Italian are not the central notion for the argument we want to make, from the beginning of the paragraph, we removed the sentence which claimed that “differences in pronouns’ reference resolution do not necessarily need to be associated with typological differences between languages”. In the current form, the paragraph (p.9 l. 189-206) does not differentiate between typological differences between the languages and the differences in sensitivity to syntactico-semantic cues. We believe that it solves the problem of circularity of the argument.

- Thematic roles are defined as the semantic roles (agent, patient,…) (see p. 6, l. 109-111). However, further on in the text, I do not understand the formulations: “the role of the thematic topic” (p. 40, l. 945) “and the thematic role of a topic” (p.41, l. 967). I would change to “the role of topic”. Furthermore, the authors propose to distinguish the syntacto-semantic cues from discourse-related cues but in the conclusion, they seem to include topicality in the syntacto-semantic cues (p. 41, l. 977) when I understood that they were talking about the discourse topic throughout the paper. This should be clarified.

Response 1.2: Thank you for spotting these inconsistencies. 

We have corrected the formulations related to the role of topic on p. 42 l. 1000 and on p. 42 l. 1010. In both instances, “the thematic topic” / “thematic role of a topic” have been replaced with “the role of topic”, following your suggestion.

We have also reworded the conclusions so that they no longer claim that topicality is an instance of a syntactico-semantic cue (which was a mistake on our part). In the current formulation the fragment in question was changed to the following form:

“Our results indicate that in Polish, syntactico-semantic cues - such as subjecthood and information structure (i.e., order of mention and thematic roles) - are the dominant constraints of the prominence of an antecedent”

- In the discussion (p. 37-38), the authors discuss the role of the context of a sentence but again, the context was not manipulated in the current study. So, contrary to what the authors claim, I do not think that the online results allow them “to determine whether these preferences can be modulated by the context of a given sentence” (p. 37, l. 858). Just as I do not think that they can conclude that null pronouns “easily shift their reference towards a more contextually appropriate referent” (p. 38, l. 888) since null pronouns, whether disambiguated in favour of the object or the subject, were presented in the same contexts.

Response 1.3: Thank you for bringing up this unfortunate use of the term “context” . We agree with the Reviewer that as context was not manipulated in our study we cannot draw any conclusions related to how it influences the pronoun resolution. Whenever we used the term “context” we actually referred to the grammatical constraints of a sentence, which were manipulated in Study 2. We replaced all the mentions of “context” in the discussion with “grammatical constraints”.

The problematic formulations were changed in the text to the following:

“[...] to determine whether these preferences can be modulated by the grammatical constraints of a given sentence”

“[...] null pronouns are forced to refer to an antecedent which does not follow the preferred antecedent match, they can easily shift their reference towards a less-preferred, but grammatically correct referent”

- In further discussion (p. 41-42), the argument about the anti-focus effect needs clarification. To the best of my knowledge, contrary to what the authors claim (p. 40, l. 952-953), an increased processing cost associated with an anti-focus effect was not observed. The anti-focus effect is the fact that the pronoun refers preferentially to the topic of the current discourse unit (which is not the focused by cleft antecedent of the discourse unit in which it appears since the clefting function is to signal a topic shift for the subsequent discourse unit). The authors should clarify what they want to argue here.

Response 1.4: Thank you for spotting this inconsistency. It has been corrected: the section that mentioned it is supposed to refer, in a broader sense, to experiments that explored the focus effect in pronoun resolution. We do not aim to explain the anti-focus effect as it is not central to our argumentation so we have removed the mention of it. The corrected version of the paragraph is the following:

“Previous studies on focus effect in anaphora resolution reported a similar effect: an increased processing cost associated with a topic shift towards a less-prominent antecedent which has been cued by the information structure of the current discourse (10,11,14,15). In this context, our results can be interpreted as showing that Polish speakers automatically interpret the most-prominent antecedent (the first available antecedent or the subject antecedent) to be the topic of a sentence, but as new information cues another antecedent which is coherent with the current discourse, the reference is shifted.”

Smaller suggestions:

- p. 2, l. 19: in the abstract, change “we explored the natural biases that occur during the interpretation of null or overt pronouns” to “we explored the interpretation preferences of null and overt pronouns”

Response 1.5: This fragment has been changed according to the Reviewer’s suggestion.

- p. 9, l. 196, change “pronoun resolution seem” to “pronoun resolution seems”

Response 1.6: Thank you for spotting this mistake. It has been corrected in the text.

- p. 10, l. 235, change “the natural pronoun-antecedent match” to “the preferred pronoun-antecedent match”

 Response 1.7: We changed this formulation according to the Reviewer’s suggestion.

- p. 11, l. 247, change “the natural anaphora resolution preference” to “the preferred anaphora interpretation”

 Response 1.8: We changed this formulation according to the Reviewer’s suggestion. 

- p. 11, l. 255, why is “followed” in italics?

 Response 1.9: The format has been changed to a regular text.

- p. 11, l. 256, change “a violation of the natural preference” to “a violation of the offline preference”

 Response 1.10: We changed this formulation according to the Reviewer’s suggestion. 

- p. 13, l. 294, change “neither the preference” to “neither the preferences”

Response 1.11: Thank you for spotting this issue. It has been corrected in the text.

- p. 14, 332, in the title, change “What is the natural pattern of anaphora resolution in Native Polish speakers?” to “What is the offline pattern of anaphora resolution in Native Polish speakers?”

Response 1.12: We changed this formulation according to the Reviewer’s suggestion. 

- p. 16, l. 366-367, change “In order to test whether a pronoun can influence the natural tendency to interpret the antecedent of a sentence” to “In order to test whether the pronoun form (null vs overt) can influence its interpretation preference”

Response 1.13: This fragment has been changed according to the Reviewer’s suggestion.

- p. 20, l. 477, change “violation of the natural preference for anaphora resolution” to “violation of the preferred anaphora interpretation” 

Response 1.14: We changed this formulation according to the Reviewer’s suggestion. 

- p. 27, l. 635, change “on native Polish speakers’ natural preference” to on native Polish speakers’ offline preference”

Response 1.15: We changed this formulation according to the Reviewer’s suggestion. 

- p. 33, l. 758, change “the natural pattern of pronominal anaphora resolution in Polish” to “the pattern of pronominal anaphora resolution in Polish”

Response 1.16: We changed this formulation according to the Reviewer’s suggestion. 

- p. 33, l. 759, change “we established a preference” to “we established the offline preference"

Response 1.17: We changed this formulation according to the Reviewer’s suggestion. 

- p. 33, l. 768, change “the interpretation of a null pronoun is not disrupted by a violation of its preference” to “the interpretation of a null pronoun is not disrupted by a violation of its subject-preference”

Response 1.18: We changed this formulation according to the Reviewer’s suggestion. 

- p. 34, l. 777-779, change “Polish speakers interpret sentences under the working assumption that the topic of a sentence is the most-prominent, first-mentioned, subject antecedent” to “Polish speakers interpret sentences under the working assumption that the most-prominent referent is the first-mentioned, subject and topic antecedent”

Response 1.19: This fragment has been changed according to the Reviewer’s suggestion.

- I did not find Appendix A

Response 1.20: We have previously added the Appendix to the submission as a separate file - we were not aware that the Reviewers will not be able to access them. It has been fixed and the Appendix is not added at the end of the manuscript. 

Reviewer #2

I think that this paper has improved substantially since the previous version, I appreciate the efforts to which the authors have gone to address all comments in the last round. My remaining comments are quite minor. The area that now requires a bit of refinement is the discussion, as you will see from my final few comments.

Page 4 line 69: this example needs a number and a translation

Response 2.1: The number has been added to the example. Translation is provided in the gloss in the line following the example sentence.

Page 4 line 74: I wouldn’t describe this as a mismatch between a pronoun and a verb. It is a bit confusing to describe it this way, although I know what you are getting at. Rephrase. I think this issue is made more confusing by the fact that, up to this point in the manuscript, you haven’t indicated exactly what information the pronoun carries in Polish (also this is not clear from the example gloss). This should be clarified.

Response 2.2: Thank you for pointing out this issue. The problematic formulation has been replaced with the following one:

“A consequence of subject-verb agreement is that whenever a pronoun of the verb phrase refers to a less-preferred, non-topical or unexpected antecedent, a speaker needs to adjust the interpretation of the sentence and re-identify the referent of the anaphorical expression.”

We hope that in this corrected, more descriptive form this fragment is not confusing anymore.

Also, we have updated the gloss of the example sentence (1) so that it explains what information is carried by the pronoun.

Page 10, lines 217: note that the SMASH algorithm is what Kehler and colleagues are arguing **against**

See also page 38 line 880.

Response 2.3: Thank you for spotting this, it was indeed our misinterpretation of Kehler and colleagues’ claims. As the discussion of the SMASH algorithm is not central to explaining the ideas proposed by Kehler and Rhode, we decided that the mention of SMASH can be omitted altogether in the text without substantially affecting the argumentation. It was also not discussed in detail in the Discussion, where it was only mentioned as an example of a certain type of pronoun reference theories (the mentioned p. 38 line 880). As such, we believe that both mentions of the SMASH algorithm of pronoun resolution can be dropped from the text without causing substantial changes to the narration.

Page 15, line 343: I think there is some kind of paste/text error here

Response 2.4: Thank you for spotting this, it has been fixed.

Page 15, line 352-354. Without knowing the contents of the Alonso-Ovalle or Chamorro studies, it is very unclear why sentences with null pronouns should be overall more natural than those containing overt pronouns. Explain.

Response 2.5: We clarified what the previous studies found and how those findings affect our predictions. In the current version of the text the relevant fragment is the following:

“Moreover, following the studies of Alonso-Ovalle and colleagues (28) and Chamorro and colleagues (21) who found null pronoun sentences are interpreted as more natural than overt pronoun sentences. Following their results, we expected to observe a similar pattern of results in our data.”

Page 16, examples: number the examples, and give a translation as well (it is normal to present both a gloss and a translation)

Response 2.6: Numbers and translations have been added to the text.

Page 19, statistical reporting: check if the term “b coefficient” is the correct one for this type of statistical analysis. I think what you normally get from a Bayesian ordinal model is an effect size and a credible interval. Check that you are reporting this correctly.

Response 2.7: We would like to thank the Reviewer #2 for this comment. We have reviewed once again the way of reporting the results of the bayesian ordinal models.

The “b coefficient” that we reported in the previous version of the text corresponds to the “Estimate” column from the brms model summary which represents the posterior means (or medians, in the case of the post-hoc tests) of the parameters of the reported bayesian regression mode (“Ordinal Regression Models in Psychology: A Tutorial” by Bürkner and Vuorre, 2018). The β coefficients are usually reported in experiments which use bayesian ordinal models, e.g.:

• Zimmerman et al. (2021): https://www.frontiersin.org/articles/10.3389/fpsyg.2021.675633/full#h4

• Skovgaard-Olsen et al. (2019): https://www.sciencedirect.com/science/article/abs/pii/S0010028518301749

• Robin et al (2021): https://doi.org/10.1080/09658211.2021.1895221

However, the convention is usually to refer to the reported estimates using the “β” notation instead of “b coefficient” so we have adjusted the text by replacing the “b coefficient” with “β”. We have also replaced the reference to credible intervals “CI” with “CrI”, which is less ambiguous (as in frequentist statistics “CI” refers to the confidence intervals). 

Page 19, line 446-447: “subject-matching sentences were rated as more natural than object-matching sentences for sentences containing overt pronouns. But in Table 1, the mean rating for subjects is much lower than for objects.

Response 2.8: Thank you for spotting this inconsistency. We have corrected the description of the results which is now the following:

“Direct comparisons between the two Antecedent conditions (i.e., subject- and object-matching sentences) revealed significant effects for sentences containing a null pronoun (b coefficient = -0.42, 95% HPD [-0.62 -0.20]): subject-matching sentences were rated as more natural than object-matching sentences. Significant effects were also found for sentences containing overt pronouns (b coefficient = 0.56, 95% HPD [0.29 0.81]); in the case of which, subject-matching sentences were rated as less natural than object-matching sentences. “

Page 20, study 1 summary: add some nuance to this paragraph. It sounds here as though you are describing categorical patterns (null pronoun only interpreted as subject and overt only as object), which is not the case.

Response 2.9: Thank you for bringing this issue to our attention. We rephrased the summarizing sentence to stress that it is not a categorical pattern but rather a preference. In the current form it is as follows:

“The results of Study 1 showed that Polish native speakers prefer to interpret the null pronoun of a subordinate clause as referring to the subject of the main clause and the overt pronoun of a subordinate clause as referring to the object of the main clause.”

Page 25: provide a translation for the example materials

Response 2.10: Similarly as in the case of previous example sentences, numbers and translations have been added to the text.

Page 32, line 724: “sentences referring to object antecedent were rated on average as more natural than sentences referring to the subject antecedent”. You can ignore this main effect of antecedent here, as it interacts with pronoun - so the main effect of antecedent is driven entirely by the difference in the object pronouns. (In general, if main effects are qualified by a significant interaction they should not be interpreted on their own)

Response 2.11: Thank you for this comment. We agree that the main effect itself is not informative in this case so we have removed the sentence referring to the main effect of Antecedent from the text. Instead, we only talk about the effect of Antecedent in the context of its interaction with the effect of Pronoun so the current version of the description of the results of the naturalness rating in Study 2 is the following:

“We found a strong effect of Pronoun (β = -0.40; 95% CrI [-0.69 -0.10]): sentences containing null pronouns were rated as more natural than sentences containing overt pronouns. We also found evidence for the interaction of Pronoun and Antecedent effects (β = -1.12, 95% CrI [-1.39 -0.84]). Direct comparisons between the two Antecedent conditions (i.e. subject- and object-matching sentences) revealed that there was no evidence for differences in naturalness score in sentences containing a null pronoun (β = 0.15, 95% HPD [-0.10 0.42]) but there was a strong effect in sentences containing overt pronouns (β = 1.27, 95% HPD [0.99 1.52]): subject-matching sentences were rated as less natural than object-matching sentences.”

Discussion section: I suggest condensing this, it is quite long and in some places repeats the same point in several ways.

Response 2.12: We have shortened the discussion by removing a couple of fragments which repeated the same information and presented similar points. We have also removed the fragment related to the discussion of the spillover effect and lexical access, as suggested by the Reviewer in the following comment. We hope that it is now more condensed and to the point.

Pages 39-40: spillover effect and lexical access. It is difficult to see how this relates directly to the current study. The studies you cite are talking about why effects show up at the spillover region at all (as opposed to a region containing a particular stimulus). But the question you need to answer is why your spillover effect goes in a particular direction. I think this is an entirely different question, so I suggest cutting out this short part of the discussion and references 49-51.

Response 2.13: Thank you for this comment. Following your suggestion we have deleted the fragment discussing the spillover effect in the context of lexical access and the corresponding references.

Page 40: you should not make it sound as though Patterson & Felser were the first to propose that pronoun processing occurs in stages over time, this idea has been around in the literature for a very long time

Response 2.14: Thank you for bringing our attention to this issue. We have reworded the sentence in which we previously referred to the work of Patterson & Felser to make it clear that we would like to refer to a broader idea. Instead of referring to Patterson & Felser we added a reference to Kehler (2007) who summarizes and critically analyzes the idea of incremental pronoun resolution:

“This explanation is in line with the idea that pronoun resolution is a process that unfolds over time (for a discussion see: Kehler, 2007): in its early stages, the initial bonding or retrieval are reflected by early fixation measures, but late processes related to the integration with discourse only affect later processing stages.”

Pages 40-41: I think you have misunderstood the anti-focus effect: it does not represent a cost associated with a “less prominent, non-focused antecedent”

Response 2.15: Thank you for spotting this mistake. We decided to remove the mention of the anti-focus effect as the section that mentioned it is supposed to refer, in a broader sense, to experiments that explored the focus effects in pronoun resolution. The corrected version of the paragraph is the following:

“Previous studies on focus effect in anaphora resolution reported a similar effect: an increased processing cost associated with a topic shift towards a less-prominent antecedent which has been cued by the information structure of the current discourse (10,11,14,15). In this context, our results can be interpreted as showing that Polish speakers automatically interpret the most-prominent antecedent (the first available antecedent or the subject antecedent) to be the topic of a sentence; however, as new information cues another antecedent which is coherent with the current discourse, the reference is shifted.”

---

## [Editor Report · Decision Letter 2]

22 Nov 2021

PONE-D-21-12740R2Pronominal anaphora resolution in Polish: investigating online sentence interpretation using eye-tracking.PLOS ONE

Dear Dr. Wolna,

Thank you for submitting your manuscript to PLOS ONE. After careful consideration, we feel that it has merit but does not fully meet PLOS ONE’s publication criteria as it currently stands. Therefore, we invite you to submit a revised version of the manuscript that addresses the points raised during the review process.

While I think the reviewer comments have been adequately addressed, your manuscript still contains some minor errors, omissions and unclear wordings (see below for details). Please attend to these issues before submitting a final revised version of your manuscript.

We look forward to receiving your revised manuscript.

Kind regards,

Claudia Felser, Ph.D

Academic Editor

PLOS ONE 

Journal Requirements:

Additional Editor Comments:

General points

1. The manuscript still contains several grammatical errors and should be carefully proofread before final submission. Examples found on the first few pages include:

Abstract: "sensitivity to violation" > "sensitivity to a violation"

80: "the reference" > "reference"

2. In-text references should be in angular rather than round brackets, following the usual convention and to avoid literature references being confused with references to linguistic examples.

Specific points

Abstract: I'm sorry for being picky, but the phrase "ambiguous sentence interpretation task" is still ambiguous as I pointed out before, in that the adjective "ambiguous" can be understood as modifying either "sentence" or "task". Why not simply omit this adjective?

68: Examples are only numbered from here, and the first two examples on p. 3 are not numbered. Please use example numbering consistently.

158: The EPP is not direction-specific and thus does not require the subject to be preverbal. Remove "preverbal" or refer explicitly to Polish, as appropriate.

307 (and elsewhere): Your use of the term "study" is confusing. You report two experiments here which both form part of the same study. Please replace "study" by "experiment" thoughout the manuscript when referring to your experiments.

350: More information about the participants is needed - minimally, their age should be provided.

380: "Qualtrics"? Please provide the URL to this platform or other reference.

p.24: Consider using a symbol other than an asterisk to indicate dispreference as asterisks normally indicate ungrammaticality.

533: Again, more participant bio-information is needed.

571f.: If this is a footnote or endnote, please format and position it as such.

p.25: How long did the experiment take?

775 (and elsewhere): Why does the word "Studies" start with a capital letter?

833ff. "In the second study of the current paper we explored to what extent the pronoun resolution preferences can be modulated by the grammatical constraints of a given sentence. Our results indicate that sentences containing a null pronoun equally easily accommodate verb phrases referring to subject and object antecedents." - It is not clear what is meant by "grammatical constraints" here (a term which replaces the previously used but misleading term "context"). Do you mean a sentence's grammatical properties, its structure, or something else? Secondly, how can verb phrases possibly refer to subject or object antecedents? Please rephrase.

Fig.1: Why does "Noun Phrase" start with capitals?
---

## [Author Response · Author response to Decision Letter 2]

8 Dec 2021

We would like to thank the Editor for her comments and suggestions. We have addressed them all and we have had the manuscript proofread by a native English speaker to make sure that there are no grammatical errors left. We hope that the revisions in the manuscript and our accompanying responses successfully address the concerns of the Editor and that after revision you will find our work suitable for publication in PLOS One.

Yours sincerely,

Agata Wolna, Joanna Durlik, Zofia Wodniecka

Response to the Editor’s comments:

Journal Requirements:

Response: The reference list had not changed since the last round of reviews and we previously checked that it was complete and that the status of the cited work was correct.

General points

1. The manuscript still contains several grammatical errors and should be carefully proofread before final submission. Examples found on the first few pages include:

Abstract: "sensitivity to violation" > "sensitivity to a violation"

80: "the reference" > "reference"

Response E.1. The specific examples you gave have been changed according to your suggestions. Moreover, the entire text has been proofread by a professional proofreader who is a native English speaker to make sure all other errors were corrected.

2. In-text references should be in angular rather than round brackets, following the usual convention and to avoid literature references being confused with references to linguistic examples.

Response E.2. We have updated the literature references in the manuscript and they are now all between angular brackets.

Specific points

Abstract: I'm sorry for being picky, but the phrase "ambiguous sentence interpretation task" is still ambiguous as I pointed out before, in that the adjective "ambiguous" can be understood as modifying either "sentence" or "task". Why not simply omit this adjective?

Response E.3. Thank you for pointing out this inconsistency. We would rather keep the adjective as it refers to one of the most crucial differences between stimuli used in experiments 1 and 2; however, we have reformulated the problematic sentence and we hope it is clear now. In the corrected version, this fragment is now as follows:

“First, we explored preferences in interpretation of null and overt pronouns in ambiguous sentences.”

68: Examples are only numbered from here, and the first two examples on p. 3 are not numbered. Please use example numbering consistently.

Response E.4. Thank you for spotting this inconsistency. Numbers have now been added for the first two examples, and the numeration of subsequent examples throughout the text has been updated accordingly.

158: The EPP is not direction-specific and thus does not require the subject to be preverbal. Remove "preverbal" or refer explicitly to Polish, as appropriate.

Response E.5. Thank you for pointing this out. The sentence has been changed according to your suggestion and it now reads as follows:

“The precise type of the expression in a Spec IP position is further defined by the Extended Projection Principle (EPP), according to which the Spec IP of a given clause must be occupied by a subject [20], which in the case of Polish will always be a preverbal subject” (p. 8 lines 161-163).

307 (and elsewhere): Your use of the term "study" is confusing. You report two experiments here which both form part of the same study. Please replace "study" by "experiment" thoughout the manuscript when referring to your experiments.

Response E.6. The terminology used in the manuscript has been updated following your suggestion: whenever we refer to one of our experiments, we now use the term “experiment”. The term “study” is used only in reference to previous studies (conducted by someone else).

350: More information about the participants is needed - minimally, their age should be provided.

Response E.9. We added the information on participants’ age to the manuscript (p. 15 line 358). In this study we only collected the information on participants’ gender, age and native language, so we do not have any more precise information describing the group in this case.

380: “Qualtrics”? Please provide the URL to this platform or other reference.

Response E.8. We now provide the URL to the platform in the manuscript: “The experiment was conducted via an online survey platform (Qualtrics: www.qualtrics.com).”

p.24: Consider using a symbol other than an asterisk to indicate dispreference as asterisks normally indicate ungrammaticality.

Response E.9. Thank you for bringing this to our attention. We replaced asterisks with “?” to mark dispreference in sentence interpretation in the text.

533: Again, more participant bio-information is needed.

Response E.10. In the case of Experiment 2, more information on the participants’ group was provided but only after explaining the exclusion criteria. However, we agree that it was not the best way of providing information. We moved the fragment with the information on the final sample size along with the demographic information to the beginning of the paragraph (p. 23, lines 545-548), and we only discuss the exclusion criteria after giving this information. In the case of Experiment 2, in addition to the information on gender and age, we also provide information on participants’ language background in Table 2 (p. 24 line 569). We only collected precise language background information on participants of Experiment 2 and not Experiment 1 because in the case of Experiment 2 we wanted to be sure that our participants would not be functional and very proficient bilinguals. The reason for restricting the participants’ sample to functional monolinguals was driven by previous findings which show that language experience (immersion in L2 or high proficiency levels) can affect online sentence interpretation and pronoun resolution measured with eye-tracking.

571f.: If this is a footnote or endnote, please format and position it as such.

Response E.11. As PLOS one does not allow us to add footnotes to the manuscript, we have moved this information to the main text (p. 24 lines 584-586), directly preceding the example of a sentence’s structure. The fragment explaining the structure of experimental sentences is now as follows:

“The length of the main clause, as well as the frequency of nouns corresponding to the subject and object antecedents, were also matched between sentences. Note that if a sentence included a null pronoun (which is equivalent to omitting the pronoun), no empty space or other replacement were provided in place of the pronoun in the sentence. Each sentence followed the same structure:

subject / verb1 / object / time conjunction (“when”) / pronoun (null* or overt) / verb2 / rest1 / rest2”

p.25: How long did the experiment take?

Response E.12. The experiment took approximately 45 minutes. We have included this information in the text (p. 26, lines 633-634).

775 (and elsewhere): Why does the word "Studies" start with a capital letter?

Response E.13. First, following one of your previous comments, we replaced all occurrences of “study” with “experiment”. Second, when talking about experiments, we use capital letters in some cases to specifically refer to our experiments (i.e., “Experiment 1” / “Experiment 2”). As such, these expressions are used as if they were proper nouns, in contrast to “experiment(s)”, which is used as a common noun. Therefore, we would rather not change the words starting with capital letters to lowercase so as to keep this distinction in the text. However, if you have a strong preference for this change, we will be happy to adopt it.

833ff. "In the second study of the current paper we explored to what extent the pronoun resolution preferences can be modulated by the grammatical constraints of a given sentence. Our results indicate that sentences containing a null pronoun equally easily accommodate verb phrases referring to subject and object antecedents." - It is not clear what is meant by "grammatical constraints" here (a term which replaces the previously used but misleading term "context"). Do you mean a sentence's grammatical properties, its structure, or something else? Secondly, how can verb phrases possibly refer to subject or object antecedents? Please rephrase.

Response E.14. We have rephrased this fragment to make it more specific. We have added a sentence to explain what we mean by “grammatical constraints'' and we have replaced the term “verb phrase” with “verb”. After corrections, this fragment reads as follows:

“In Experiment 2, we explored the extent to which pronoun resolution preferences can be modulated by the grammatical constraints of a given sentence. In the case of Polish, grammatical constraints refer to grammatical number, person or gender agreement between the verb and its antecedent. Our results indicate that sentences containing a null pronoun accommodate verbs referring to subject and object antecedents equally easily.” (p. 36, lines 858-863)

Fig.1: Why does "Noun Phrase" start with capitals?

Response E.15. There was no real reason for that. We have corrected the text in the figure and it no longer starts with capital letters.

---

## [Editor Report · Decision Letter 3]

13 Dec 2021

PONE-D-21-12740R3Pronominal anaphora resolution in Polish: investigating online sentence interpretation using eye-tracking.PLOS ONE

Dear Dr. Wolna,

Thank you for submitting your manuscript to PLOS ONE. After careful consideration, we feel that it has merit but does not fully meet PLOS ONE’s publication criteria as it currently stands. Therefore, we invite you to submit a revised version of the manuscript that addresses the points raised during the review process.

We look forward to receiving your revised manuscript.

Kind regards,

Claudia Felser, Ph.D

Academic Editor

PLOS ONE

Journal Requirements:

Additional Editor Comments (if provided):

You have addressed most of my previous comments well enough, but please check your manuscript again carefully for grammatical and linguistic accuracy. Reading through your current manuscript I spotted the following two points, but there may be more.

214: on one hand > on the one hand

840ff.: “In Experiment 2, we explored the extent to which pronoun resolution preferences can be modulated by the grammatical constraints of a given sentence. In the case of Polish, grammatical constraints refer to grammatical number, person or gender agreement between the verb and its antecedent. Our results indicate that sentences containing a null pronoun accommodate verbs referring to subject and object antecedents equally easily.” - The reworded passage makes no sense. Verbs cannot refer to subject or object antecedents any more than verb phrases can! It is pronouns that require antecedents. The constraints you mention are not constraints of sentences - I am guessing that what you mean are constraints on referential dependency formation. Please reword as appropriate.
---

## [Author Response · Author response to Decision Letter 3]

21 Dec 2021

We have carefully read through the manuscript and fixed a number of grammatical and linguistic inaccuracies and mistakes. We have also had the manuscript proofread by a native English speaker to make sure that there are no grammatical errors left. We hope that the revisions in the manuscript successfully address the concerns of the Editor and that after revision you will find our work suitable for publication in PLOS One.

Yours sincerely,

Agata Wolna, Joanna Durlik and Zofia Wodniecka

Response to the Editor’s comments:

You have addressed most of my previous comments well enough, but please check your manuscript again carefully for grammatical and linguistic accuracy. Reading through your current manuscript I spotted the following two points, but there may be more.

Thank you for pointing out to this problem. We have carefully read through the manuscript and fixed a number of grammatical and linguistic inaccuracies. We have also asked a professional proofreader (a native English speaker) to read the manuscript again and we have introduced the corrections he has suggested.

214: on one hand > on the one hand

This mistake has been corrected following the Editor’s suggestion.

840ff.: “In Experiment 2, we explored the extent to which pronoun resolution preferences can be modulated by the grammatical constraints of a given sentence. In the case of Polish, grammatical constraints refer to grammatical number, person or gender agreement between the verb and its antecedent. Our results indicate that sentences containing a null pronoun accommodate verbs referring to subject and object antecedents equally easily.” - The reworded passage makes no sense. Verbs cannot refer to subject or object antecedents any more than verb phrases can! It is pronouns that require antecedents. The constraints you mention are not constraints of sentences - I am guessing that what you mean are constraints on referential dependency formation. Please reword as appropriate.

We reformulated this fragment following the Editor’s suggestion. It is now as follows:

“In Experiment 2, we explored the extent to which pronoun resolution preferences can be modulated by the grammatical constraints on the formation of a referential dependency. In the case of Polish, these constraints refer to the grammatical number, person, or gender agreement between the pronoun and its antecedent. Our results indicate that null pronouns can equally easily be interpreted as referring to subject and object antecedents.” (p.36, lines 843 - 847)

---

## [Editor Report · Decision Letter 4]

24 Dec 2021

Pronominal anaphora resolution in Polish: investigating online sentence interpretation using eye-tracking.

PONE-D-21-12740R4

Dear Dr. Wolna,

We’re pleased to inform you that your manuscript has been judged scientifically suitable for publication and will be formally accepted for publication once it meets all outstanding technical requirements.

Kind regards,

Claudia Felser, Ph.D

Academic Editor

PLOS ONE
---

## [Editor Report · Acceptance letter]

3 Jan 2022

PONE-D-21-12740R4 

Pronominal anaphora resolution in Polish: investigating online sentence interpretation using eye-tracking. 

Dear Dr. Wolna:

I'm pleased to inform you that your manuscript has been deemed suitable for publication in PLOS ONE. Congratulations! Your manuscript is now with our production department. 

Kind regards, 

on behalf of

Dr. Claudia Felser 

Academic Editor

PLOS ONE